# Stochastic bandits with groups of similar arms

**Fabien Pesquerel**[*]
fabien.pesquerel@inria.fr

**Hassan Saber**
hassan.saber@inria.fr

**Odalric-Ambrym Maillard**
odalric.maillard@inria.fr

Univ. Lille, CNRS, Inria, Centrale Lille, UMR 9198-CRIStAL, F-59000 Lille, France

## Abstract

We consider a variant of the stochastic multi-armed bandit problem where arms are known to be organized into different groups having the same mean. The groups are unknown but a lower bound $q$ on their size is known. This situation typically appears when each arm can be described with a list of categorical attributes, and the (unknown) mean reward function only depends on a subset of them, the others being redundant. In this case, $q$ is linked naturally to the number of attributes considered redundant, and the number of categories of each attribute. For this structured problem of practical relevance, we first derive the asymptotic regret lower bound and corresponding constrained optimization problem. They reveal the achievable regret can be substantially reduced when compared to the unstructured setup, possibly by a factor $q$. However, solving exactly the exact constrained optimization problem involves a combinatorial problem. We introduce a lower-bound inspired strategy involving a computationally efficient relaxation that is based on a sorting mechanism. We further prove it achieves a lower bound close to the optimal one up to a controlled factor, and achieves an asymptotic regret $q$ times smaller than the unstructured one. We believe this shows it is a valuable strategy for the practitioner. Last, we illustrate the performance of the considered strategy on numerical experiments involving a large number of arms.

## 1 Introduction

The finite stochastic multi-armed bandit problem is a popular framework for studying sequential decision making problems in which a learner sequentially samples from a finite set of distributions called arms. It was first introduced in the context of medical trials [Thompson, 1933b, 1935] and later formalized under this name by Robbins in Robbins [1952]. We refer the interested reader to Lattimore and Szepesvári [2020] for a recent survey. This is one of the simplest theoretical framework in which one can study the notion of *exploration-exploitation* tradeoff. This tension between *exploration* and *exploitation* arises from the sequential optimization problem a learner is trying to perform while being uncertain about the very problem it is optimizing.

Formally, a multi-armed bandit configuration is specified by a set of unknown real-valued probability distributions $\nu = (\nu_a)_{a \in \mathcal{A}}$ with means $(\mu_a)_{a \in \mathcal{A}}$, indexed by a set of arms $\mathcal{A}$. We hereafter consider a finite $\mathcal{A}$, and that all $\nu_a$, $a \in \mathcal{A}$ belong to the same family of distributions $\mathcal{F}$ (e.g. Bernoulli, Gaussian, etc.), that is $\nu \in \mathcal{F}^{\mathcal{A}}$. The bandit game proceeds as follows. At each time $t \in \mathbb{N}$, the learner chooses an arm $a_t \in \mathcal{A}$ based on the past observations and decisions, then receives and observes a sample $X_t$ (called the reward), conditionally independent, sampled from $\nu_{a_t}$. Her goal is to maximize the cumulative reward received over time. The mean of each arm is unknown, which makes the problem non-trivial, hence the learner should adjust her sampling strategy based on past information obtained from drawing different arms in order to maximize the expected sum of rewards. The maximal expected value of a finite bandit configuration is denoted by $\mu_\star$, defined as $\mu_\star = \max_{a \in \mathcal{A}} \mu_a$. The performance of the strategy used by the agent is measured by the (pseudo) *regret*, that compares the

35th Conference on Neural Information Processing Systems (NeurIPS 2021).

expected sum of rewards obtained by an oracle that would constantly pull an optimal arm and the ones obtained by the learner, up to some time horizon $T$ (that we assume is unknown to the learner).

**Definition 1** (Regret). *The regret incurred by a sampling strategy after $T$ time steps on a bandit configuration $\nu$ is given by:*

$$\mathcal{R}\left(\nu, T\right) = \mathbb{E}_\nu \left( \sum_{t=1}^T \left(\mu_* - \mu_{a_t}\right) \right) = \sum_{a \in \mathcal{A}} \left(\mu_* - \mu_a\right) \mathbb{E}_\nu \left(N_a(T)\right),$$

*where $N_a(T) = \sum_{t=1}^T \mathbb{I}\{a_t = a\}$ denotes the number of selection of arm $a$ after $T$ time steps.*

**Group of similar arms** Motivated by various practical reasons, one may want to restrict to a subset $\mathcal{B} \subset \mathcal{F}^{\mathcal{A}}$ of allowed bandit configurations instead of the full set $\mathcal{F}^{\mathcal{A}}$. In this paper, we study a variant of the multi-armed bandit problem in which the reward function, $\mu : a \in \mathcal{A} \to \mu_a$, is assumed to satisfy a cluster-like structural property. A bandit configuration $\nu$ is said to satisfy the **q-equivalence property** if for every arm $a \in \mathcal{A}$, there are at least $q-1$ distinct arms having the same expected value:

$$\forall a \in \mathcal{A}, \quad |\{a' \in \mathcal{A} | \mu_{a'} = \mu_a\}| \geqslant q.$$

Assuming the set of arms $\mathcal{A}$ and base distributions $\mathcal{D}$ are known to the learner, we denote by $\mathcal{B}_q$ the set of bandit configurations having the q-equivalence property. We also denote by $\mathcal{B}_q(\mu)$ the set of all expected values in $\mathcal{B}_q$. Formally, $\mathcal{B}_q(\mu)$ is the image of $\mathcal{B}_q$ under the $\mu$ mapping.

**Definition 2** (Arm equivalence and equivalence class). *Given a bandit configuration $\nu$, two arms $a, a' \in \mathcal{A}$ are said to be equivalent if their associated distributions have the same expected values:*

$$a \sim a' \Leftrightarrow \mu_a = \mu_{a'}.$$

*An equivalence class $c$ in $\nu$ is a maximal subset of arms in $\mathcal{A}$ having the same mean, i.e., for all arms $a, a'$ in $c$, $\mu_a = \mu_{a'}$ and for all arm $a \in c$ and $a' \in \mathcal{A} \setminus c$, $\mu_a \neq \mu_{a'}$.*

This situation typically appears in practical situations when each arm can be described with a list of categorical attributes, and the (unknown) mean reward function only depends on a subset of them, the others being redundant. In this case, $q$ is naturally linked to the number of attributes considered redundant (or useless descriptors), and the number of categories of each attribute. Precisely, $q = \prod_{i \in \mathcal{R}} c_i$ where $\mathcal{R}$ is the set of redundant attributes and $c_i$ the number of categories for attribute $i$. The learner may know that there exists such a structure while not knowing a closed form formula mapping the list of categorical attributes to the significant subset. In this case, $q$ might be a lower bound on the sizes of the class since the set $\mathcal{R}$ might not be the largest possible one or because the number of redundant attributes depends on the number of relevant attributes. In all cases, the smallest possible number of redundant attributes can be naturally linked to $q$. We hereafter consider the learner only knows $q$ but would like to exploit the prior knowledge of this structure in a bandit problem.

**Regret lower bounds overview** In order to assess the performance of a bandit algorithm on a set of configurations $\mathcal{B}$, one naturally studies the best guarantee achievable by a uniformly efficient algorithm on $\mathcal{B}$, i.e with sub-linear regret on any instance $\nu \in \mathcal{B}$ of the bandit problem. When $\mathcal{B} = \mathcal{F}^{\mathcal{A}}$, such a guarantee was first provided by Lai and Robbins [1985] for parametric families $\mathcal{F}$, and then extended by Burnetas and Katehakis [1996] for more general families. It states that any algorithm that is uniformly efficient[1] on a family of distributions $\mathcal{F}$ must satisfy

$$\liminf_{T \to \infty} \frac{\mathcal{R}(\nu, T)}{\log(T)} \geqslant \sum_{a : \mu_\star > \mu_a 0} \frac{\mu_\star - \mu_a}{\mathcal{K}_{\mathcal{F}}(\nu_a \| \mu^\star)}, \quad \mathcal{K}_{\mathcal{F}}(\nu_a \| \mu^\star) = \inf_{G \in \mathcal{F}} \{\mathrm{KL}(\nu_a \| G) : \mathbf{E}_G(X) > \mu^\star\} . \quad (1)$$

This popular result entails that any strategy having the desirable property to have sub-linear regret on any instance in $\mathcal{F}$ must incur a non-trivial minimal regret. When $\mathcal{B}$ is a strict subset of $\mathcal{F}^{\mathcal{A}}$, the bandit problem is called structured, as in this case pulling an arm may reveal information that makes it possible to refine estimation of other arms (e.g. think of the set of bandit configurations having Lipschitz mean function with respect to $\mathcal{A} \subset \mathbb{R}^d$). The presence of structure may considerably modify the achievable lowest regret, as shown in Burnetas and Katehakis [1996] and Graves and Lai

---

[1]Formally, for each bandit on $\mathcal{F}$, for each arm $k$ with $\Delta_k > 0$, then $\mathbf{E}[N_k(T)] = o(T^\alpha)$ for all $\alpha \in (0, 1]$.

[1997], who extended the (unstructured) lower bounds to arbitrarily structured bandit problems (and beyond). These lower bound take the generic form $\liminf_{T \to \infty} \dfrac{\mathcal{R}(\nu, T)}{\log(T)} \geqslant \mathfrak{C}_{\mathcal{B}}(\nu)$, where $\mathfrak{C}_{\mathcal{B}}(\nu)$ is a constant term solution of a constrained linear-optimization problem. A bandit algorithm is then called *asymptotically optimal* for a set $\mathcal{B}$ when its regret asymptotically matches this lower bound.

**Existing strategies**  In order to minimize the regret, a learner faces the classical exploration/exploitation dilemma: it needs to balance *exploration*, that is gaining information about the expected values of the arms by sampling them, and *exploitation*, that is playing the most promising arm sufficiently often. Many algorithms have been proposed to solve the multi-armed bandits problem (see Lattimore and Szepesvári [2020] for a survey). The study of the lower bounds had a crucial impact on the development of provably asymptotically optimal strategies. In the case of *unstructured* bandit $\mathcal{B} = \mathcal{F}^{\mathcal{A}}$, this includes strategies that build on the concept of *Optimism in Face of Uncertainty* (the most celebrated of which being the Upper Confidence Bound (UCB) algorithms Agrawal [1995], Auer et al. [2002]), such as KLUCB [Lai, 1987, Cappé et al., 2013, Maillard, 2018], DMED and IMED Honda and Takemura [2011, 2015], that are proven asymptotically optimal for various families $\mathcal{F}$ (e.g. one-dimensional exponential families), and directly exploit the lower bound in their structure. Alternative asymptotically optimal strategies include the Thompson Sampling (TS) Thompson [1933a], Agrawal and Goyal [2012], which uses a Bayesian posterior distribution given a specific prior, whose optimality was shown in Korda et al. [2013]. See also Kveton et al. [2019] for other randomized algorithms and Kveton et al. [2020], Chan [2020], Baudry et al. [2020] for recent non-parametric extensions using re-sampling methods. Further, some authors also allow many optimal arms, see de Heide et al. [2021], or even countably many arms, see Kalvit and Zeevi [2020]. However, these works do not consider nor exploit a constraint on the level-sets of the mean function and follow an optimistic paradigm while we follow an information minimization targeting optimality. On the other hand, several instances of structured bandits received considerable attention in the last few years. This is the case for instance of linear bandits, see [Abbasi-Yadkori et al., 2011, Srinivas et al., 2010, Durand et al., 2017, Kveton et al., 2020] and Lattimore and Szepesvari [2017], Lipschitz bandits Magureanu et al. [2014], Wang et al. [2020], Lu et al. [2019], unimodal bandits Yu and Mannor [2011], Combes and Proutiere [2014], Saber et al. [2020], or combinatorial bandits Kveton et al. [2015], Magureanu [2018], and more recently Cuvelier et al. [2021b]. A generic asymptotically optimal algorithm, called OSSB (Optimal Structured Stochastic Bandit), has been introduced in the work of Combes et al. [2017], and proven to be asymptotically optimal for all structures satisfying some weak properties that include all the aforementioned structures. Although being asymptotically optimal this algorithm often suffers from a long burn-in phase that may hinder its finite practical performance. It further comes with high computational price as it requires to solve an empirical version of the optimization problem $\mathfrak{C}_{\mathcal{B}}(\nu)$ at each step. This motivates the quest for alternative strategies, perhaps less generic but better suited to a specific structure. Inspired by combinatorial structures for which computing $\mathfrak{C}_{\mathcal{D}}(\nu)$ is simply not feasible, a relaxation of the generic constrained optimization problem was recently proposed in Cuvelier et al. [2021a]. The authors show that this comes at the price of trading-off regret optimality for computational efficiency. Indeed in some structure, combinatorial properties are at stake and asymptotically optimal algorithms may require solving combinatorial optimization problems (see Cuvelier et al. [2021a]) related to $\mathfrak{C}_{\mathcal{B}}(\nu)$. In order to exploit the combinatorial structures in a numerically efficient way, research has been made in how to relax these combinatorial optimization problems while preserving theoretical properties on the regret of the relaxed algorithms (see Cuvelier et al. [2021b,a]). Our work consider similar computational issues, with a different perspective.

**Goal**  For the structure $\mathcal{B}_q$, as we show in Theorem 1 below, the term $\mathfrak{C}_{\mathcal{B}_q}(\nu)$ unfortunately makes appear in general a combinatorial optimization problem. This makes resorting to OSSB or any strategy targeting exact asymptotic optimality a daunting task for the practitioner. In this paper, our goal is to provide a computationally efficient strategy adapted to the structure $\mathcal{B}_q$, that is able to reach optimality up to controlled error term.

**Outline and contributions**  The rest of this paper is organized as follows. In section 2, we derive a lower bound on the regret for the structured set of bandit configurations $\mathcal{B}_q$. This bound makes appear two components, one that we call *non-combinatorial* as optimizing it can be done efficiently, and a second term that we term *combinatorial* as it involves solving a combinatorial problem. Interestingly, using in Lemma 1 and Theorem 3 that the contribution of the combinatorial part of the lower bound

can be controlled. Owing to this key insight, we introduce in section 3, `IMED-EC`, an adaptation of the IMED strategy from Honda and Takemura [2015] to the structured set $\mathcal{B}_q$. One advantage of IMED over a KL-UCB alternative is its reduced complexity, which translates to the equivalence class setup. At each time step, the complexity of computing the next arm to be pulled by `IMED-EC` is no more than the one of sorting a list of $|\mathcal{A}|$ elements once the IMED indexes have been computed, which is only $\log |A|$ times larger than looking for the minimal IMED index. In Section 4, we prove that `IMED-EC` achieves a controlled asymptotic regret that matches the non-combinatorial part of the lower bound and is at most (less than) a factor of 2 times the optimal regret bound. Last, we illustrate the benefit of the `IMED-EC` over its unstructured version in section 5, where it shows a substantial improvement. Our experiments also highlights the robustness of the algorithm to a misspecified parameter $q$, which is a desirable feature for the practitioner.

## 2    A regret lower bound with combinatorial and non-combinatorial parts

In this section, we derive a lower bound on the number of pulls of suboptimal arms that involves a combinatorial optimization problem. Using that lower bound, we derive a simple algorithm, `IMED-EC`, that does not involve any optimization problem. While not being asymptotically optimal, we will show in the next section that our algorithm has an upper bound on its regret that is no more than a fraction of the unstructured regret.

The proof of Theorem 1 is based on the concept of **most confusing instance**. Most confusing instances allow to assess the intrinsic difficulty of a bandit problem and allow to compute lower bounds on the number of times suboptimal arms are pulled. The lower bound informs us on the minimal amount of exploration one needs to do to solve a bandit problem. More formally, a confusing instance $\nu'$ associated to a suboptimal arm $a$ for a bandit problem $\nu$ is a bandit instance with the same set of arms as the original one, but in which $\mu_a$ has been changed to $\mu'_a > \mu_*$ . An optimal sampling strategy (one that does not sample suboptimal arms too much) should behave differently on the two problems. Studying this difference, we can compute the minimal amount of exploration performed by an optimal strategy on arm $a$ in the original problem $\nu$. Doing so for all suboptimal arms allows to bound the number of samples of suboptimal arms and therefore characterize the intrinsic complexity of a bandit instance $\nu$.

In a structured setting, a confusing instance also has to respect the structure. In our case, it means that a confusing instance cannot have a class with less than $q$ arms. We will therefore consider confusing instances associated to classes rather than individual arms.

**Definition 3** (Confusing instance). *Given a bandit configuration $\nu \in \mathcal{B}_q$, a real number $\lambda$ and a subset $c_q \subseteq \mathcal{A}$ of $q$ equivalent arms in $\nu$, we denote by $\mathcal{B}_q(\nu, c_q, \lambda)$ the set of all bandit configurations having the same set of arms as $\nu$ and such that for all $\nu' \in \mathcal{B}_q(\nu, c_q, \lambda)$, $\nu' \in \mathcal{B}_q$ and for every arm $a$ in $c_q$, $\mu'_a \geqslant \lambda$.*

*When $\lambda > \mu_*$, and $c_q$ is a subset of a suboptimal class, a bandit configuration in $\mathcal{B}_q(\nu, c_q, \lambda)$ is called a **confusing instance** of $\nu$.*

*Similarly to the notation introduced above, we will use the notation $\mathcal{B}_q(\mu, c_q, \lambda)$ to specify the set of means of bandit configurations in $\mathcal{B}_q(\mu, c_q, \lambda)$.*

The aim of an asymptotic lower bound on the number of pulls of a suboptimal arm is to mathematically understand the minimal asymptotic amount of exploration an algorithm should perform.

**Assumption 1:**    The family $\mathcal{F}$ is such that for all $\kappa \in \mathcal{F}$, $\mu \mapsto \mathcal{K}_{\mathcal{F}}(\kappa \| \mu)$ and $\mu \mapsto \mathcal{K}_{eq}(\kappa \| \mu)$ are continuous, where $\mathcal{K}_{eq}(\kappa, \mu) = \inf_{G \in \mathcal{F}} \{\mathrm{KL}(\kappa, G) : \mathbf{E}_G(X) = \mu\}$ with KL being a notation for the relative entropy or Kullback-Leibler divergence.

**Assumption 2:**    The family $\mathcal{F}$ is an exponential family of dimension 1. Therefore the KL divergences are parameterized by the mean and we may write the KL as a function of the means, $\forall \kappa, \chi \in \mathcal{F}, \mathrm{KL}(\kappa \| \chi) = \mathrm{KL}(\mathbb{E}_{X \sim \kappa}(X) \| \mathbb{E}_{X \sim \chi}(X))$ (identification of the KL with its parameterization by the means).

**Theorem 1** (Asymptotic lower bound). *Let $q \in \mathbb{N}_*$ be a positive integer and $\nu \in \mathcal{B}_q$ be a bandit configuration having the $q$-equivalence property. Let $c \subset \mathcal{A}$ be a suboptimal equivalence class in $\nu$.*

*Assuming uniform consistency, for all suboptimal arms a,*

$$\forall \alpha > 0, \quad \lim_{T \to +\infty} \mathbb{E}\left(\frac{N_a(T)}{T^\alpha}\right) = 0,$$

*assuming assumption 1, we have the following asymptotic bandit dependent lower bound on the number of pulls of arms in c:*

$$\liminf_{T \to \infty} \frac{\displaystyle\min_{c_q \subseteq c} \sum_{a \in c_q} \mathbb{E}_\nu(N_a(T)) \, \mathcal{K}_\mathcal{F}(\nu_a \| \mu_*) + \inf_{\mu' \in \mathcal{B}_q(\mu, c_q, \mu_*)} \sum_{a \notin c_q} \mathbb{E}_\nu(N_a(T)) \, \mathcal{K}_{eq}(\nu_a \| \mu'_a)}{\log T} \geqslant 1 , \quad (2)$$

*where $c_q$ is any subset of c having q distincts arms within it.*

We briefly sketch how confusing instances are used in the proof of Theorem 1. We consider confusing instances in which $q$ arms from a suboptimal class $c$ are moved above the optimal one (*w.r.t.* the mean). If there are $q$ arms in the class, then there are no remaining arms to move. If there are more than $2q$ arms, then moving $q$ arms creates a reminder of size larger than $q$ meaning that the crafted confusing instance respects the equivalence structure. However, if there are between $q+1$ and $2q-1$ arms, then the reminder is of size larger than 1 but strictly smaller than $q$. The created confusing instance does not respect the equivalence structure and we have to deal with the arms in the reminder (the *infimum* of equation (2)). There are $|c|$ choose $q$ possible choices to move $q$ arms from class $c$ (the *minimum* of equation (2)). All in all, the lower bound involves a combinatorial optimization problem.

While this lower bound involves a combinatorial optimization term, one can distinguish between two regimes depending on the size of the suboptimal class. The *combinatorial regime* and the *non combinatorial regime*.

**Non-combinatorial regime**    For a suboptimal class $c$, if $|c| = q$ or $|c| \geqslant 2q$, then the lower bound reduces to

$$\liminf_{T \to \infty} \frac{\displaystyle\min_{c_q \subseteq c} \sum_{a \in c_q} \mathbb{E}_\nu(N_a(T)) \, \mathcal{K}_\mathcal{F}(\nu_a \| \mu_*)}{\log T} \geqslant 1,$$

because the reminder is of size larger than $q$ and the *infimum* from Theorem 1 disappears. Indeed, the *infimum* is always 0 as this quantity can be obtained by choosing $\mu'_a = \mu_a$ for all $a \in c \setminus c_q$. Furthermore, the minimum over all $q$-partitions of $c$ is in fact the sum of the $q$ smallest elements of $\{\mathbb{E}_\nu(N_a(T)) \, \mathcal{K}_\mathcal{F}(\nu_a \| \mu_*)\}_{a \in c}$. The search amongst all the $q$-partitions of $c$ amounts to a research of the $q$ smallest elements which is not more complex than sorting a list of $|c|$ elements. Hence, the problem is no more a combinatorial optimization one and we call this case the *non-combinatorial* regime.

**Lemma 1.** *Let $\nu \in \mathcal{B}_q$ be a bandit configuration having the q-equivalence property. Let c be a suboptimal class in the non-combinatorial regime, then, under assumption 1 and 2,*

$$\liminf_{T \to \infty} \frac{\displaystyle\sum_{a \in c} \mathbb{E}_\nu(N_a(T))}{\log T} \geqslant \frac{|c|}{q} \frac{1}{\mathcal{K}_\mathcal{F}(\nu_a \| \lambda)}. \quad (3)$$

While we do not have information about individual number of times an arm in a class has been sampled, Lemma 1 roughly tells us than on average, the lower bound on the minimal amount of exploration of an arm in a suboptimal class has been divided by $q$.

**Lemma 2.** *If all suboptimal classes are in the non-combinatorial regime, under assumption 1 and 2, the regret may be asymptotically lower bounded by*

$$\liminf_{T \to \infty} \frac{\mathcal{R}(\nu, T)}{\log T} \geqslant \frac{1}{q} \sum_{a \in \mathcal{A} \setminus \mathcal{A}_*} \frac{\mu_* - \mu_a}{\mathcal{K}_\mathcal{F}(\nu_a \| \lambda)}. \quad (4)$$

Lemma 2 informs us that in the non-combinatorial regime, the classical lower bound on the regret given by equation (1) has been divided by $q$.

**Combinatorial regime** For a suboptimal class $c$ to be in the *combinatorial* regime, we need $q < |c| < 2q$, since the reminder is such that $0 < |c \smallsetminus c_q| < q$ and the infimum in Theorem 1 is not 0. In that case, the lower bound (2) involves a combinatorial optimization problem. Two difficulties arise from the term

$$\inf_{\mu' \in \mathcal{B}_q(\mu, c_q, \lambda)} \sum_{a \notin c_q} \mathbb{E}_\nu \left( N_a(T) \right) \mathcal{K}_{eq} \left( \nu_a \| \mu'_a \right).$$

First, while we could have thought that summing on the reminder $c \smallsetminus c_q$ would be enough, the summand has to be on $a \notin c_q$ as a whole. Indeed, the residual $c \smallsetminus c_q$ may be of size $q - 1$ meaning that it might cost less to move an arm from another class to the residual in order to complete it rather than moving all the reminder. Second, while we could have thought that moving elements from one class of $\nu$ to another might be enough, the *infimum* has to be taken on $\mathcal{B}_q \left( \mu, c_q, \lambda \right)$. Indeed, the residual $c \smallsetminus c_q$ may be of size $q - 1$ and the *nearest* class might be of size exactly $q$. In this case, it may cost less to move all the $2q - 1$ distributions in between the two classes and create a new one rather than merging one of the two with the other.

**Lemma 3.** *Let $\nu \in \mathcal{B}_q$ be a bandit configuration having the q-equivalence property and $c$ be a suboptimal class in the combinatorial regime. Then, under assumptions 1 and 2,*

$$\liminf_{T \to \infty} \frac{\sum_{a \in c} \mathbb{E}_\nu \left( N_a(T) \right)}{\log T} \geqslant \frac{1}{\frac{q}{|c|} \mathcal{K}_{\mathcal{F}} \left( \nu_a \| \mu_* \right) + \frac{|c| - q}{|c|} \min_{\kappa \in \nu} \mathcal{K}_{eq} \left( \nu_a \| \kappa \right)}, \tag{5}$$

$$\liminf_{T \to \infty} \frac{\sum_{a \in c} \mathbb{E}_\nu \left( N_a(T) \right)}{\log T} \geqslant \frac{1}{2q} \sum_{a \in c} \frac{1}{\mathcal{K}_{\mathcal{F}} \left( \nu_a \| \mu_* \right)}. \tag{6}$$

Those equations can be compared to the equation (3) from the non-combinatorial regime. We emphasize the fact that the lower bounds given by equations (5) and (6) are not the *largest* possible lower bound and hence do not provide as much information about the algorithmically achievable regret as the largest one given by equation (2). However, together with a regret upper bound on the algorithm `IMED-EC`, those quantities will help us control the asymptotic discrepancy between `IMED-EC`'s regret and the asymptotic lower bound given by Theorem 1.

## 3 Information Minimization for bandits with equivalence class

The algorithm we present, `IMED-EC`, depends on the (*weak*) indexes introduced in the `IMED` paper by Honda and Takemura [2015]. At each time step $t$, for each arm $a \in \mathcal{A}$, we can compute its IMED index as

$$I_a(t) = N_a(t) \mathcal{K}_{\mathcal{F}} \left( \widehat{\mu}_a(t) \| \widehat{\mu}^*(t) \right) + \log N_a(t),$$

where $\widehat{\mu}^*(t) = \max_{a \in \mathcal{A}} \widehat{\mu}_a(t)$ and for each arm $a \in \mathcal{A}$, $\widehat{\mu}_a(t)$ is the empirical mean of arm $a$ computed with samples from this arm collected up to time $t$, $\widehat{\mu}_a(t) = \frac{1}{N_a(t)} \sum_{s=1}^{t} X_s \mathbb{1} \{ a_s = a \}$, where $X_s$ is the sample collected by the algorithm at time step $s$. Let $\nu \in \mathcal{B}_q$ be a bandit configuration having the q-equivalence property. We denote by $\mathcal{A}_*(t) = \arg\max_{a \in \mathcal{A}} \{ \widehat{\mu}_a(t) \}$ the set of empirical optimal arms at time $t$. We will denote by $\mathcal{A}_q(t)$ the set of arms having the $q$ smallest IMED indexes (breaking ties randomly so that this set has size $q$). We will also consider the two following quantities for each time $t$:

$$I^*(t) = \min_{a \in \mathcal{A}_*(t)} I_a(t) = \min_{a \in \mathcal{A}_*(t)} \log N_a(t),$$

$$I(t) = \min_{\substack{\mathcal{A}' \subset \mathcal{A} \\ |\mathcal{A}'| = q}} \sum_{a' \in \mathcal{A}'} I_{a'}(t) = \sum_{a' \in \mathcal{A}_q(t)} I_a(t).$$

$I(t)$ can be computed efficiently by summing the $q$ smallest elements of the list of IMED indexes. Finding the $q$ smallest elements can be done by maintaining a sorted array of IMED indexes while computing them. The procedure costs a constant factor of $\log |\mathcal{A}|$. Computing $I(t)$ therefore costs $\mathcal{O} \left( |\mathcal{A}| \log |A| \right)$, which is only $\log |A|$ times larger than looking for the minimal IMED index. Computing $|\mathcal{A}_*(t)|$ can be done by maintaining a set of arms having the best empirical mean (adds a constant factor). The `IMED-EC` algorithm is presented in Algorithm 1.

**Algorithm 1** `IMED-EC` (IMED for Equivalent Classes)

---
Pull each arm once
**for** $t = |\mathcal{A}| \ldots T - 1$ **do**
   **if** $I^*(t) \leqslant I(t)$ **then**
      Pull $a_{t+1} \in \arg\min_{a \in \mathcal{A}_*(t)} N_a(t)$ (chosen arbitrarily)
   **else**
      Pull $a_{t+1} \in \arg\min_{a \notin \mathcal{A}_*(t)} I_a(t)$ (chosen arbitrarily)
   **end if**
**end for**

---

While the orginal problem involves combinatorial quantities, those are not involved in the `IMED-EC` algorithm. From a time complexity viewpoint, this makes this algorithm on par with other popular algorithms such as `UCB`, `KLUCB`, and `IMED` algorithm. On the contrary, the general structure algorithm `OSSB` involves solving a combinatorial optimization problem at each time step, which makes it numerically inefficient. We are not aware of any general relaxation method for this algorithm that we could compare `IMED-EC` with. It is interesting to note that in the case where $q = 1$, the `IMED-EC` algorithms coincide with the `IMED` algorithm.

**Intuition** For an arm $a$, $N_a(t)\mathcal{K}_{\mathcal{F}}(\widehat{\mu}_a(t)\|\widehat{\mu}^*(t))$ may be interpreted as the opposite of a *log-likelihood of optimality* of that arm. $\log N_a(t)$ is linked to the log-frequency of play of that arm, the frequency of play of an arm being interpreted as the probability of pulling that arm is a sequence of length $t$. The IMED algorithm thus can be intuitively understood as an algorithm matching an empirical log-probability with a log-frequency of play. In our setting, there is at least $q$ elements in each group. It therefore makes sense to test for the optimality of a group rather single elements. Since all arms are independent, it makes sense to sum the *log-likelihood of optimality* on all the q-partitions of the set of arms. Since we have the intuition that this first part is the logarithm of a product of probability, we may compare it to the product of the frequencies. Therefore, we get that important quantities are the sum of IMED indexes for each $q$ partition of the arms, seen as a comparison between the optimality of this group of $q$ elements and the associated frequency of play of that group. The minimal IMED index is the one whose frequency of play is the lowest compared to its *likelihood of optimality*, similarly for the sum of IMED indexes. Other intuitions regarding the fairness (frequency of pulls within the same class) of the algorithm are given in appendix D.

## 4 Regret analysis

In this section, we now detail the main bound on the regret of `IMED-EC`.

**Theorem 2** (Upper bound on the number of pulls). *Under the `IMED-EC` algorithms, under assumption 1 and 2, the number of pulls of a suboptimal arm $a$ is upper bounded by:*

$$\mathbb{E}_{\nu}\left(N_a(T)\right) \leqslant \frac{\log T}{q\mathcal{K}_{\mathcal{F}}\left(\nu_a\|\mu_*\right)}\left(1 + \alpha(\varepsilon)\right) + f(\varepsilon), \tag{7}$$

*where $0 < \varepsilon < \frac{1}{3}\min_{a \in \mathcal{A} \setminus \mathcal{A}_*}(\mu_* - \mu_a)$, $f$ is function that depends on concentration properties on $\mathcal{F}$, and $\alpha$ tends to 0 as $\varepsilon$ tends to 0.*

Remark: $\alpha$ and $f$ functions are mostly used for deriving theoretical guarantees in IMED-EC regret analysis. $\alpha$ is controlled thanks to property 2 as in the paper of Honda and Takemura [2015] for `IMED` regret analysis. A finite sample analysis can be derived from a careful analysis of the term $f$. Being more precise requires scrutinizing the properties of the considered family.

**Corollary 1.** *Under the `IMED-EC` algorithms, under assumptions 1 and 2, the number of pulls of a suboptimal arm $a$ is upper bounded by:*

$$\min_{c_q \subseteq c}\sum_{a \in c_q}\mathbb{E}_{\nu}\left(N_a(T)\right)\mathcal{K}_{\mathcal{F}}\left(\nu_a\|\mu_*\right) \leqslant \left(1 + \alpha\left(\varepsilon\right)\right)\log T + g(\varepsilon). \tag{8}$$

*where $0 < \varepsilon < \frac{1}{3}\min_{a \in \mathcal{A} \setminus \mathcal{A}_*}(\mu_* - \mu_a)$ $\alpha$ and $\alpha$ tends to 0 as $\varepsilon$ tends to 0.*

**Theorem 3** (Asymptotic upper bound on the number of pulls). *Under the `IMED-EC` algorithms, under assumption 1 and 2, the number of pulls of a suboptimal arm a is asymptotically upper bounded by:*

$$\liminf_{t \to +\infty} \frac{\mathbb{E}_\nu\left(N_a(T)\right)}{\log T} \leqslant \frac{1}{q\mathcal{K}_\mathcal{F}\left(\nu_a \| \mu_*\right)}. \tag{9}$$

**Discussion** This upper bound shows that in particular, the number of pulls of a suboptimal class, $\sum_{a \in c} \mathbb{E}_\nu\left(N_a(T)\right)$ is asymptotically no more than $\frac{|c|}{q\mathcal{K}_\mathcal{F}(\nu_a \| \mu_*)} \log T$. This hence matches the lower bound in the *non-combinatorial regime*. In the *combinatorial regime*, along with equation (6), this regret upper bound shows that

$$\frac{|c|}{q\mathcal{K}_\mathcal{F}\left(\nu_a \| \mu_*\right)} \geqslant \liminf_{T \to \infty} \sum_{a \in c} \frac{\mathbb{E}_\nu\left(N_a(T)\right)}{\log T} \geqslant \frac{1}{2} \cdot \frac{|c|}{q\mathcal{K}_\mathcal{F}\left(\nu_a \| \mu_*\right)},$$

proving that the regret of the proposed `IMED-EC` does not differ from the optimal lower bound by a factor more than 2. This is a striking result. Equation (6) can be used to have an even more precise control on the discrepancy to the optimal regret bound, as it shows the factor 2 can be actually replaced with $1 + \frac{|c|-q}{q} \frac{\min_{\kappa \in \nu} \mathcal{K}_{eq}(\nu_a \| \kappa)}{\mathcal{K}_\mathcal{F}(\nu_a \| \mu_*)}$. Since the factor $\frac{|c|-q}{q} \frac{\min_{\kappa \in \nu} \mathcal{K}_{eq}(\nu_a \| \kappa)}{\mathcal{K}_\mathcal{F}(\nu_a \| \mu_*)}$ is strictly between 0 and 1 in the combinatorial regime that we are studying, the discrepancy between the lower bound and the regret of `IMED-EC` indeed is always bounded by 2. On the other hand, this refined error measurement is problem dependant while the factor of 2 is universal.

We provide the full proof of Theorem 3 and Theorem 2 in appendix C where we also discuss how to weaken assumption 2 and still get the result of theorem 2.

## 5 Experiments

In this section, we support our theoretical analysis by conducting three sets of experiments. The Python code used to perform those experiments is available on Github[2]. We support our empirical evidences using plots of cumulative regrets. In this section, all the experiments are conducted using gaussian distributions whose means are between 0 and 1 and of unit standard deviation. Those graphs are representative of all the experiments that we conducted and more plots and experiments may be found in the appendix D.

**Balanced class, perfect knowledge** In this set of experiments, see Figure 1, we focus on the bandit configurations in which all equivalence classes have the same cardinality and assume that we know the number of elements per class. This setting is interesting for two reasons. First, one can compute the theoretical lowerbound without solving a combinatorial optimization problem. Second, the theoretical analysis shows that `IMED-EC` is asymptotically optimal in this case. This setting will thus allow us to numerically grasp what happens in the most structured case. We compare `IMED-EC` to unspecialized bandit algorithm, `UCB`, `IMED` and `KLUCB`. To make the comparison fairer we also compare `IMED-EC` to `OSSB`, an algorithm specialized in structured bandit. Since `OSSB` has to solve a combinatorial optimization problem at each time step, we cannot carry experiments on large sets of arms while comparing `IMED-EC` to it. In this particular setting, we see that while `OSSB` and `IMED-EC` are provably asymptotically optimal, `IMED-EC` numerically performs better in finite time horizon. We recall that it is furthermore numerically more efficient since it does not involve any combinatorial optimization. Without too much surprises, `IMED-EC` also outperforms unspecialized algorithm.

**Imperfect knowledge** In the experiment plotted Figure 2, we leverage the knowledge hypothesis and assume that we only know a lower bound on the number of elements per class while the classes are still balanced. We compare `IMED-EC` to unspecialized bandit algorithm, `IMED` and `KLUCB`. We drop `OSSB` from our test bed due to the computational burden of solving a combinatorial optimization problem at each time step. We can see that the finite time cumulative regret of `IMED-EC` indeed is much smaller than the regret of the unspecialized algorithms.

**Influence of the parameter $q$** Here we show the numerical robustness of `IMED-EC` with respect to the lower bound parameter $q$ on the number of elements per classes. On the same bandit problem,

---

[2]https://github.com/fabienpesquerel/stochastic-bandits-with-groups-of-similar-arms-neurips-2021

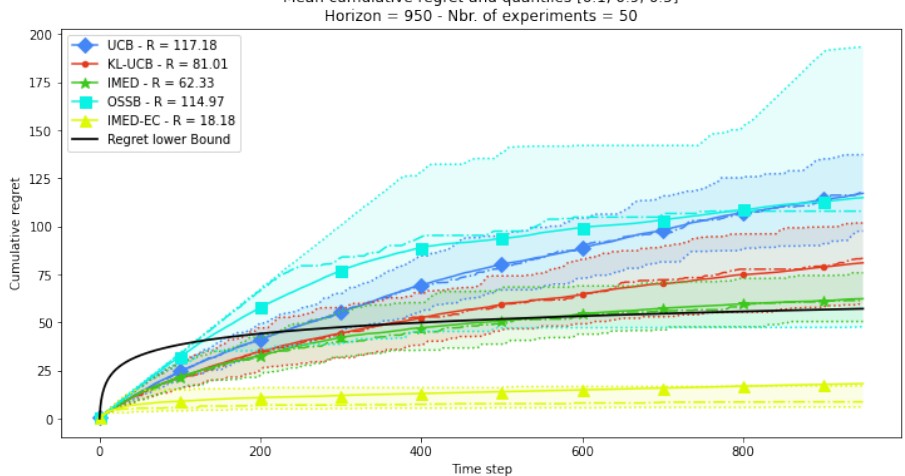

Figure 1: 3 classes, 3 distributions per class - set of means = $\{0.3, 0.5, 0.9\}$

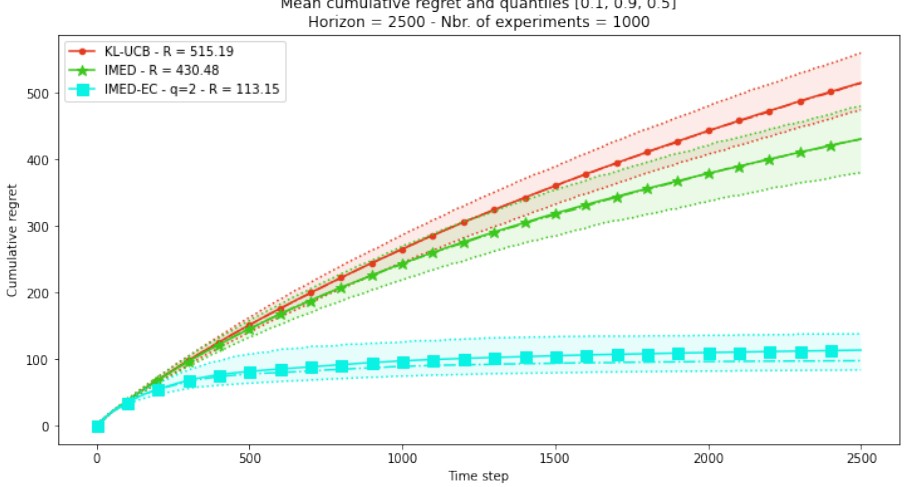

Figure 2: 7 classes, 8 distributions per class - set of means = $\{0.1, 0.3, 0.4, 0.5, 0.6, 0.75, 0.9\}$

we compare different instances of `IMED-EC` where different values of $q$ are used. In the legend, *opt.* stands for optimal and corresponds to the largest valid lower bound on the number of elements per class, *i.e.* the minimal number of elements in a class. The experiment Figure 3 is performed on a bandit problem with 7 classes and an uneven number of distributions per class. The smallest class has 4 elements and the largest 23. While $q$ increases up to the minimum cardinality of a class, we see that the performances of `IMED-EC` increases. It is rather remarkable that once we go beyond that theoretical threshold, the performances of `IMED-EC` do not deteriorate. We even found it difficult to find settings to deteriorate them at all. While the expected regret does not seem to deteriorate, we sometimes see that the tails of the regret widen as it can be seen on the plot Figure 3 for $q = 7$ and $q = 20$ since the 0.9 quantile curves are so large for those values of $q$. We interpret part of this robustness to the fact that the relaxation induced in `IMED-EC` makes the algorithm over explore compared to what the true lower bound suggests. Increasing $q$ reduces the exploration and therefore may improve the performances of the algorithm. However, this robustness is observed even in the case where the classes are balanced. This interpretation thus does not explain everything about the numerical robustness of `IMED-EC`. This type of experiment does not take more than roughly 10 to 15 minutes on a notebook run in Google Colab depending on the number of arms, the horizon and the number of runs. This supports the numerical efficiency of the relaxation made in `IMED-EC`.

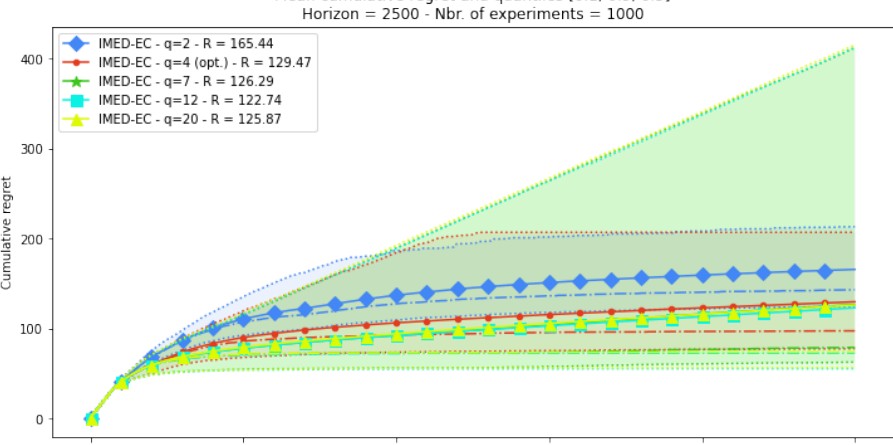

Figure 3: 7 classes, unbalanced - set of means = $\{0.1, 0.3, 0.4, 0.5, 0.6, 0.75, 0.9\}$

## 6 Conclusion

In this paper, we introduced `IMED-EC`, a numerically efficient algorithm to solve a structured bandit problem for which we derived a lower bound involving a combinatorial optimization problem. While not being asymptotically optimal, we proved that the asymptotic regret of `IMED-EC` is always smaller than the unstructured one and that we can control the discrepancy with respect to the structured regret lower bound by a factor of at most 2.

## Acknowledgments and Disclosure of Funding

This work has been supported by the French Ministry of Higher Education and Research, Inria, Scool, the French Agence Nationale de la Recherche (ANR) under grant ANR-16-CE40-0002 (the BADASS project) the MEL and the I-Site ULNE regarding project R-PILOTE-19-004-APPRENF. The PhD of Fabien Pesquerel is supported by a grant from École Normale Supérieure.

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
