# A  Notations and Assumptions

In this section, we first recall a few notations. Then, we provide details about the assumptions made about the distributions. In particular, we make precise the assumption about exponential families we consider, and we also provide an alternative set of assumptions under which our analysis hold. Indeed, our proof techniques naturally apply to the setup considered for the analysis of the IMED strategy. We formalize the corresponding assumptions below in (Assumption 1, 3).

**Notations:**  Given a bandit configuration $\nu$ and an arm $a \in \mathcal{A}$, $\widehat{\mu}_a(T) = \frac{1}{N_a(T)} \sum_{t=1}^{T} X_t \mathbb{1}\{a_t = a\}$ where $N_a(T) \sum_{t=1}^{T} \mathbb{1}\{a_t = a\}$ and $X_t$ is the reward obtained at time $t$. We will denote by $\widehat{\mu}_a^n$ the empirical mean of arm $a$ obtained from $n$ *i.i.d.* samples (and $n$ is not a random variable).

**Assumption 1:**  The family $\mathcal{F}$ is such that for all $\kappa \in \mathcal{F}$, $\mu \mapsto \mathcal{K}_\mathcal{F}(\kappa\|\mu)$ and $\mu \mapsto \mathcal{K}_{eq}(\kappa\|\mu)$ are continuous, where $\mathcal{K}_{eq}(\kappa\|\mu) = \inf_{G \in \mathcal{F}}\{\mathrm{KL}(\kappa\|G) : \mathbf{E}_G(X) = \mu\}$ with KL being a notation for the relative entropy or Kullback-Leibler divergence.

This property is also assumed in Honda and Takemura [2015]. This is especially relevant as they consider reward distributions having a semi-bounded support.

**Assumption 2:**  The family $\mathcal{F}$ is an exponential family of dimension 1, *i.e.* admits the canonical decomposition with respect to some measure

$$p_\theta(x) = \exp\left(t(x)\theta - \psi(\theta) + k(x)\right),$$

where $\theta \in \Theta \subseteq \mathbb{R}$ is a parameter, $k$ a real function, $t$ is called the sufficient statistics and $\psi$, the log-partition function, is assumed to be twice differentiable. We assume that $\Theta$ is open and non-empty. We further assume that on $\Theta$, the second derivative, $\psi''$, of the log-partition function is bounded. Formally, there exists $M_{\psi,\Theta}$ such that:

$$\sup_{\theta \in \Theta} \psi''(\theta) \leqslant M_{\psi,\Theta}. \tag{10}$$

If $p_\theta$ and $p_{\theta'}$ are two distributions in $\mathcal{F}$, then:

$$\mathrm{KL}(p_\theta\|p_{\theta'}) = \psi(\theta) - \psi(\theta') - (\theta - \theta')\psi'(\theta').$$

Because $\mathcal{F}$ is an exponential family of dimension 1, $\psi'' > 0$ and $\psi'$ is strictly increasing. This implies that there is a continuous bijection between the parameter space $\Theta$ and the space of expected values that can be taken between distributions in $\mathcal{F}$. Specifically, $\mu : \theta \mapsto \mathbb{E}_{X \sim p_\theta}(X)$ is a bijection on its co-domain, $\mu(\Theta)$. Therefore the KL divergences can be parameterized by the mean and we may write the KL as a function of the means, $\forall \kappa, \chi \in \mathcal{F}$, $\mathrm{KL}(\kappa\|\chi) = d(\mathbb{E}_{X \sim \kappa}(X)\|\mathbb{E}_{X \sim \chi}(X))$.

Therefore, under assumption 2 $\mathcal{K}_\mathcal{F}$ identifies with the KL and assumption 1 holds by twice differentiability of $\psi$ since it implies the continuity of $\psi$ and $\psi'$. Note that exponential families of dimension 1 have been considered in several other works, see e.g. Korda et al. [2013], Cappé et al. [2013], or Maillard [2018] and Lai [1988] where restriction on $\psi''$ is also considered.

**Assumption 3:**  The family $\mathcal{F}$ is the set of distributions with semi-bounded rewards, *i.e.* whose supports lies in $(-\infty, 1]$. Furthermore, the moment generating function $\mathbb{E}(\exp(\lambda X))$ of any distribution $X \in \mathcal{F}$ exists in a neighborhood of $\lambda = 0$. For technical reasons linked to the moment generating function, we also assume that the maximal expected rewards of the distributions in $\mathcal{F}$ is strictly smaller than 1. Those assumptions on $\mathcal{F}$ are the same than the one of the paper Honda and Takemura [2015].

Under assumption 2, the moment generating function can be written as a function of $\psi$,

$$\mathbb{E}_{X \sim p_\theta}(\exp(\lambda X)) = \exp(\psi(\theta + \lambda) - \psi(\theta)),$$

which is well defined as long as $\theta + \lambda \in \Theta$. This is the reason assumption 1 consider $\Theta$ to be *open* since in this case, $\theta$ can never be too close to the boundary, *i.e.* there always exists $\lambda$ small enough such that $\theta + \lambda \in \Theta$, and the moment generating function always exists in a neighborhood of 0.

**Remark:** The regret lower bound can be proved under assumption 2 or alternatively under assumption 1 and 3. Most of the distributions studied in the bandit literature fall under one of those sets of assumptions. Assumption 2 includes for instance Gaussian bandits (known variance) and Bernoulli distributions. For Bernoulli distributions, the set of considered means should be of the form $(\varepsilon, 1 - \delta)$ with $0 < \varepsilon < 1 - \delta < 1$ in order to upper bound the second derivative of the log-partition function. This situation is represented in Figure 4. Now for Gaussian distributions with known variance, (10) holds trivially.

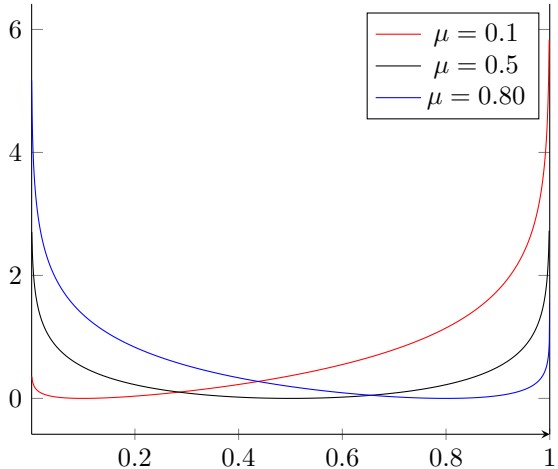

Figure 4: Plot of $x \mapsto \mathrm{kl}\left(\mu \| x\right) = x \log \frac{x}{\mu} + (1 - x) \frac{1-x}{1-\mu}$ (Bernoulli)

**Property 1** (Monotonicity of the KL). *Under assumption 2, if we assume that,*

$$a \leqslant \mu + \varepsilon < \mu_* - \varepsilon \leqslant b,$$

*then,*

$$d\left(a\|b\right) \geqslant d\left(\mu + \varepsilon \| \mu_* - \varepsilon\right).$$

The paper of Honda and Takemura [2015] features a similar property for the $\mathcal{K}_{\mathcal{F}}$ under assumption 1 and assumption 3.

**Property 2** (Upper approximation of the KL). *Let $\varepsilon$ be such that $\mu + \varepsilon < x - \varepsilon$. Under assumption 2, there exists $\alpha$ such that*

$$\frac{1}{d\left(\mu + \varepsilon \| x - \varepsilon\right)} \leqslant \frac{1}{d\left(\mu \| x\right)} \left(1 + \alpha\left(\varepsilon\right)\right), \tag{11}$$

*with $\alpha(\varepsilon) \to 0$ as $\varepsilon \to 0$.*

The paper of Honda and Takemura [2015] features a similar property for the $\mathcal{K}_{\mathcal{F}}$ under assumption 1 and assumption 3.

**Proposition 1.** *Let $\mathcal{F}$ be an exponential family $\mathcal{F}$ of dimension 1 satisfying the hypothesis of assumption 2. Let $X \sim p_{\theta_*} \in \mathcal{F}$ be a real random variable. Denote by $\theta_*$ its true natural parameter and $\widehat{\theta}_n$ it empirical parameter computed with $n$ i.i.d. samples from the distribution of $X$. Let $\psi$ be the log-partition function of the family $\mathcal{F}$. Let $\theta$ be the parameter corresponding to the distribution of $\mathcal{F}$ with mean $\mu < \mu_*$ and $\theta_*$ be the parameter corresponding to the disribution of $\mathcal{F}$ with mean $\mu_*$, then*

$$\mathbb{P}\left(\mathrm{KL}\left(p_{\widehat{\theta}_n}\|p_\theta\right) \geqslant u, \widehat{\mu}_n \leqslant \mu\right) \leqslant \mathbb{P}\left(\mathrm{KL}\left(p_{\widehat{\theta}_n}\|p_{\theta_*}\right) \geqslant u + \alpha_\psi\left(\theta_*, \theta\right)\sqrt{u}\right)$$
$$\leqslant \exp\left(-n\left(u + \alpha_\psi\left(\theta_*, \theta\right)\sqrt{u}\right)\right),$$

*where $\alpha_\psi\left(\theta_*, \theta\right) > 0$.*

*Proof.* For distributions in an exponential family, the Kullback-Leibler divergence can be written using $\psi''$, and therefore:

$$\mathrm{KL}\left(p_{\widehat{\theta}_n}\|p_\theta\right) = \frac{1}{2}\left(\widehat{\theta}_n - \theta\right)^2 \psi''\left(\gamma\widehat{\theta}_n + (1 - \gamma)\theta\right) \geqslant u.$$

Under assumption 2, $\psi''$ is uniformly bounded, $\psi'' \leqslant M$, therefore:

$$\left| \theta - \widehat{\theta}_n \right| \geqslant \sqrt{\frac{2u}{M}}.$$

Without loss of generality, we can assume that

$$\theta - \widehat{\theta}_n \geqslant \sqrt{\frac{2u}{M}}. \tag{12}$$

because the following derivation will feature the term $\langle \theta - \widehat{\theta}_n | \psi'(\theta_*) - \psi'(\theta) \rangle$. This term is always positive because $\widehat{\mu}_n \leqslant \mu < \mu_*$ and $\mu : \theta \mapsto \mathbb{E}_{X \sim p_\theta}(X)$ is a continuous bijection. Therefore it has to be strictly monotonous. Since $\psi'$ is strictly increasing (because $\psi$ is strictly convex), it preserves the order. Therefore, $\theta - \widehat{\theta}_n$ and $\psi'(\theta_*) - \psi'(\theta)$ have the same sign.

Since $\mathrm{KL}\left(p_\theta \| p_{\theta'}\right) = \psi(\theta) - \psi(\theta) - (\theta - \theta')\psi'(\theta')$, we can derive the following sequence of implications:

$$\Leftrightarrow \mathrm{KL}\left(p_{\widehat{\theta}_n} \| p_\theta\right) \geqslant u$$

$$\Leftrightarrow \mathrm{KL}\left(p_{\widehat{\theta}_n} \| p_\theta\right) + \mathrm{KL}\left(p_{\widehat{\theta}_n} \| p_{\theta_*}\right) \geqslant u + \mathrm{KL}\left(p_{\widehat{\theta}_n} \| p_{\theta_*}\right)$$

$$\Leftrightarrow \mathrm{KL}\left(p_{\widehat{\theta}_n} \| p_{\theta_*}\right) \geqslant u + \underbrace{\mathrm{KL}\left(p_{\widehat{\theta}_n} \| p_{\theta_*}\right) - \mathrm{KL}\left(p_{\widehat{\theta}_n} \| p_\theta\right)}_{>0}$$

$$\Leftrightarrow \mathrm{KL}\left(p_{\widehat{\theta}_n} \| p_{\theta_*}\right) \geqslant u + \mathrm{KL}\left(p_\theta \| p_{\theta_*}\right) + \langle \theta - \widehat{\theta}_n | \underbrace{\psi'(\theta_*) - \psi'(\theta)}_{>0 \ because \ \psi \ is \ strictly \ convex} \rangle$$

$$\Leftrightarrow \mathrm{KL}\left(p_{\widehat{\theta}_n} \| p_{\theta_*}\right) \geqslant u + \mathrm{KL}\left(p_\theta \| p_{\theta_*}\right) + \langle \underbrace{\theta - \widehat{\theta}_n}_{\geqslant \sqrt{2u/M} \ by \ (12)} | g(\theta_*, \theta) \rangle$$

$$\Rightarrow \mathrm{KL}\left(p_{\widehat{\theta}_n} \| p_{\theta_*}\right) \geqslant u + \mathrm{KL}\left(p_\theta \| p_{\theta_*}\right) + \langle \sqrt{\frac{2u}{M}} | g(\theta_*, \theta) \rangle$$

$$\Rightarrow \mathrm{KL}\left(p_{\widehat{\theta}_n} \| p_{\theta_*}\right) \geqslant u + \alpha_\psi(\theta_*, \theta)\sqrt{u}.$$

The upper bound on the probability is a classical result about exponential families. $\qquad \square$

## B  Proof of the regret lower bound

In this section we prove Theorem 1 that we remind below.

**Theorem** (Asymptotic lower bound). *Let $q \in \mathbb{N}_*$ be a positive integer and $\nu \in \mathcal{B}_q$ be a bandit configuration having the q-equivalence property. Let $c \subset A$ be a suboptimal equivalence class in $\nu$. Assuming uniform consistency, for all suboptimal arm $a$,*

$$\forall \alpha > 0, \quad \lim_{T \to +\infty} \mathbb{E}\left(\frac{N_a(T)}{T^\alpha}\right) = 0,$$

*assuming assumption 1, we have the following asymptotic bandit dependent lower bound on the number of pulls of arms in c:*

$$\liminf_{T \to \infty} \frac{\min_{c_q \subseteq c} \sum_{a \in c_q} \mathbb{E}_\nu\left(N_a(T)\right) \mathcal{K}_{\mathcal{F}}\left(\nu_a \| \mu_*\right) + \inf_{\mu' \in \mathcal{B}_q(\mu, c_q, \lambda)} \sum_{a \notin c_q} \mathbb{E}_\nu\left(N_a(T)\right) \mathcal{K}_{eq}\left(\nu_a \| \mu_a'\right)}{\log T} \geqslant 1,$$

*where $c_q$ is any subset of c having q distinct arms within it.*

The proof is standard and makes use of the notion of *most confusing instance* specialized for this structure in the main part of this paper.

*Proof of Theorem 1.* Let $I_T = (a_k, X_{a_k})_{1 \leqslant k \leqslant T}$ denote the history of actions and rewards taken by a sequential decision maker algorithm up to time $T$. Then, using the data processing inequality, it is proved in Garivier et al. [2016] that

$$\sum_{a \in \mathcal{A}} \mathbb{E}_\nu \left( N_a(T) \right) \mathrm{KL} \left( \nu_a \| \nu'_a \right) \geqslant \mathrm{kl} \left( \mathbb{E}_\nu \left( Z \right) \| \mathbb{E}_{\nu'} \left( Z \right) \right),$$

for $Z \in (0,1)$ a $\sigma(I_T)$-measurable random variable and kl the Kullback-Leibler divergence between Bernoulli distributions. Let $c$ be suboptimal class and $c_q \subseteq c$ be subset of $q$ elements in $c$. Applying the previous inequality for $Z = N_c(T)/T$, for all $\lambda > \mu_*$ and $\nu' \in \mathcal{B}_q(\nu, c_q, \lambda)$ we have that:

$$\sum_{a \in c} \mathbb{E}_\nu \left( N_a(T) \right) \mathrm{KL} \left( \nu_a \| \nu'_a \right) = \sum_{a \in c_q} \mathbb{E}_\nu \left( N_a(T) \right) \mathrm{KL} \left( \nu_a \| \nu'_a \right) + \sum_{a \notin c_q} \mathbb{E}_\nu \left( N_a(T) \right) \mathrm{KL} \left( \nu_a \| \nu'_a \right)$$

$$\geqslant kl \left( \mathbb{E}_\nu \left( \frac{N_c(T)}{T} \right) \| \mathbb{E}_{\nu'} \left( \frac{N_c(T)}{T} \right) \right)$$

$$\geqslant \left( 1 - \mathbb{E}_\nu \left( \frac{N_c(T)}{T} \right) \right) \log \frac{1}{1 - \mathbb{E}_{\nu'} \left( \frac{N_c(T)}{T} \right)} - \log 2$$

$$= \left( 1 - \mathbb{E}_\nu \left( \frac{N_c(T)}{T} \right) \right) \log \frac{T}{T - \mathbb{E}_{\nu'} \left( N_c(T) \right)} - \log 2.$$

Since all arms that are not in $c$ are suboptimal for $\nu'$, the uniform consistency hypothesis implies that

$$\forall 0 < \alpha \leqslant 1, \qquad 0 \leqslant T - \mathbb{E}_{\nu'} \left( N_c(T) \right) = o \left( T^\alpha \right);$$

and therefore, $T - \mathbb{E}_{\nu'} \left( N_c(T) \right) \leqslant T^\alpha$ for $T$ large enough. We deduce that, for all $0 < \alpha \leqslant 1$,

$$\liminf_{T \to +\infty} \frac{1}{\log T} \log \frac{T}{T - \mathbb{E}_{\nu'} \left( N_c(T) \right)} \geqslant \liminf_{T \to +\infty} \frac{1}{\log T} \log \frac{T}{T^\alpha} = 1 - \alpha.$$

Since all arms within the class $c$ are suboptimal for $\nu$ and the considered strategy is assumed to satisfy the uniform consistency hypothesis, $\mathbb{E}_\nu \left( \frac{N_c(T)}{T} \right) \to 0$ as $T \to +\infty$. Together, and letting $\alpha$ be arbitrarily close to 0, these facts implies that

$$\liminf_{T \to \infty} \frac{\sum_{a \in c_q} \mathbb{E}_\nu \left( N_a(t) \right) \mathrm{KL} \left( \nu_a \| \nu'_a \right) + \sum_{a \notin c_q} \mathbb{E}_\nu \left( N_a(t) \right) \mathrm{KL} \left( \nu_a \| \nu'_a \right)}{\log T} \geqslant 1.$$

For each $c_q$, we can minimize this quantity over all confusing instances $\nu' \in \mathcal{B}_q(\nu, c_q, \lambda)$ (with a 0 lower bound if the set is empty), and use the continuity of the KL (assumption 1) to let $\lambda > \mu_*$ tends toward $\mu_*$,

$$\liminf_{T \to \infty} \frac{\sum_{a \in c_q} \mathbb{E}_\nu \left( N_a(T) \right) \mathcal{K}_\mathcal{F} \left( \nu_a \| \mu_* \right) + \inf_{\mu' \in \mathcal{B}_q(\mu, c_q, \lambda)} \sum_{a \notin c_q} \mathbb{E}_\nu \left( N_a(T) \right) \mathcal{K}_{eq} \left( \nu_a \| \mu'_a \right)}{\log T} \geqslant 1,$$

where each KL can be minimized independently once $\mu' \in \mathcal{B}_q(\mu, c_q, \lambda)$ has been set, owing to the considered structure. Since this lower bound is valid for all $c_q \in c$ suboptimal, it is valid for the minimal quantity over all $q$ partitions,

$$\liminf_{T \to \infty} \frac{\min_{c_q \subseteq c} \sum_{a \in c_q} \mathbb{E}_\nu \left( N_a(T) \right) \mathcal{K}_\mathcal{F} \left( \nu_a \| \mu_* \right) + \inf_{\mu' \in \mathcal{B}_q(\mu, c_q, \lambda)} \sum_{a \notin c_q} \mathbb{E}_\nu \left( N_a(T) \right) \mathcal{K}_{eq} \left( \nu_a \| \mu'_a \right)}{\log T} \geqslant 1,$$

which proves the Theorem 1. $\qquad \square$

# C Proof of the regret upper bound

In this section, we prove the regret upper bound, Theorem 2, incurred by the `IMED-EC` algorithm.

**Theorem** (Upper bound on the number of pulls). *Under the `IMED-EC` algorithms, under assumption 1 and 2, the number of pulls of a suboptimal arm $a$ is upper bounded by:*

$$\mathbb{E}_{\nu}\left(N_a(T)\right) \leqslant \frac{\log T}{q\mathcal{K}_{\mathcal{F}}\left(\nu_a\|\mu_*\right)}\left(1 + \alpha(\varepsilon)\right) + f(\varepsilon), \tag{13}$$

*where $0 < \varepsilon < \frac{1}{3}\min_{a\in\mathcal{A}\setminus\mathcal{A}_*}\left(\mu_* - \mu_a\right)\alpha$ and $\alpha$ tends to 0 as $\varepsilon$ tends to 0.*

The proof proceeds in several steps. We first derive *empirical* bounds on the number of pulls of a suboptimal arm given that this arm is being pulled at time $t$.

**Lemma 4** (Empirical bounds). *Let $a_{t+1}$ be the pulled arm at time $t+1$, and $a$ be any arm belonging to $\mathcal{A}_*(t)$ at some time $t$. Under the `IMED-EC` algorithm,*
*if $a_{t+1} \in \mathcal{A}_*(t)$,*

$$N_{a_{t+1}}(t) \leqslant N_a(t), \tag{14}$$

$$\log\left(N_{a_{t+1}}(t)\right) \leqslant \min_{c_q \subseteq c_*}\sum_{k\in c_q} N_k(t)d\left(\widehat{\mu}_k(t)\|\widehat{\mu}^*\right) + \log N_k(t), \tag{15}$$

*and if $a_{t+1} \notin \mathcal{A}_*(t)$,*

$$qN_{a_{t+1}}(t)d\left(\widehat{\mu}_{a_{t+1}}\|\widehat{\mu}^*\right) \leqslant \log t, \tag{16}$$

$$q\log\left(N_{a_{t+1}}(t)\right) \leqslant \min_{c_q \subseteq c_*}\sum_{k\in c_q} N_k(t)d\left(\widehat{\mu}_k(t)\|\widehat{\mu}^*\right) + \log N_k(t). \tag{17}$$

*Proof.* Assume that the chosen arm, $a_{t+1}$, belongs to $\mathcal{A}_*(t)$, then by definition of $I(t)$ and $I^*(t)$, $I^*(t) \leqslant I(t)$. $I^*(t) = I_{a_{t+1}}$ because $a_{t+1}$ is the chosen arm amongst elements of $\mathcal{A}_*(t)$, hence belongs to $\arg\min_{a\in\mathcal{A}_*(t)}\log N_a(t)$. Equation (14) follows and $N_{a_{t+1}} \leqslant N_a(t)$ for all $a \in \mathcal{A}_*(t)$. Equation (15) then follows from the fact that:

$$\begin{aligned}
\log N_{a_{t+1}}(t) &= I^*(t) \\
&\leqslant I(t) \\
&= \min_{\substack{\mathcal{A}'\subset\mathcal{A}\\|\mathcal{A}'|=q}}\sum_{a'\in\mathcal{A}'} I_{a'}(t) \\
&\leqslant \min_{c_q \subseteq c_*}\sum_{k\in c_q} N_k(t)d\left(\widehat{\mu}_k(t)\|\widehat{\mu}^*\right) + \log N_k(t).
\end{aligned}$$

Assume that the chosen arm, $a_{t+1}$, does not belong to $\mathcal{A}_*(t)$. Then $I(t) \leqslant I^*(t)$. The flow of control of the `IMED-EC` algorithm implies that $a_{t+1}$ is an arm with minimal IMED index. By definition of $I(t)$, $q$ times $I_{a_{t+1}}$ will always be smaller than or equal to $I(t)$:

$$q\left(N_{a_{t+1}}(t)\,d\left(\widehat{\mu}_{a_{t+1}}(t)\,\|\widehat{\mu}^*(t)\right) + \log N_{a_{t+1}}(t)\right) \leqslant I(t). \tag{18}$$

By definition of $I^*(t)$, there exists $a \in \mathcal{A}_*(t)$ such that $I^*(t) = \log N_a(t)$ implying that $I^*(t) \leqslant \log t$. Since $I(t) \leqslant I^*(t)$, it implies that $I(t) \leqslant \log t$. From equation (18) we deduce equation (16),

$$qN_{a_{t+1}}(t)d\left(\widehat{\mu}_{a_{t+1}}\|\widehat{\mu}^*\right) \leqslant \log t.$$

Last, equation (17) can be deduced from the definition of $I(t)$ and equation (18):

$$\begin{aligned}
q\log N_{a_{t+1}}(t) &\leqslant q\left(N_{a_{t+1}}(t)\,d\left(\widehat{\mu}_{a_{t+1}}(t)\,\|\widehat{\mu}^*(t)\right) + \log N_{a_{t+1}}(t)\right) \\
&\leqslant I(t) \\
&= \min_{\substack{\mathcal{A}'\subset\mathcal{A}\\|\mathcal{A}'|=q}}\sum_{a'\in\mathcal{A}'} I_{a'}(t) \\
&\leqslant \min_{c_q \subseteq c_*}\sum_{k\in c_q} N_k(t)d\left(\widehat{\mu}_k(t)\|\widehat{\mu}^*\right) + \log N_k(t).
\end{aligned}$$

$\square$

If we were to substitute empirical means with real ones, so that $\widehat{\mu}_a(t) = \mu_a$ for all $a \in \mathcal{A}$ and $\mathcal{A}_*(t) = \mathcal{A}_*$, then one can see that equation (16) gives us the desired behavior, *i.e.* if $a_{t+1}$ is a suboptimal arm, $N_{a_{t+1}}(t) \leqslant \frac{\log t}{qd\left(\mu_{a_{t+1}} \| \mu_*\right)}$. For distributions having *concentration around the mean* property we will be able to say that, for large enough $t$, with high probability, for all arms $a \in \mathcal{A}$, $|\widehat{\mu}_a(t) - \mu_a| \leqslant \varepsilon$. In this case, we shall still have a desired property. This is the statement of the next Lemma. We then proceed to show the concentration properties later on.

**Lemma 5.** *Let $0 < \varepsilon \leqslant \frac{1}{3} \min_{a \notin \mathcal{A}_*} (\mu_* - \mu_a)$. Assume that $a_{t+1}$ is the pulled arm at time $t + 1$ and $a_{t+1} \in \mathcal{A} \setminus \mathcal{A}_*$ is a suboptimal arm. Let's assume that $\widehat{\mu}_{a_{t+1}}(t) \leqslant \mu_a + \varepsilon$. Let's also assume that $\widehat{\mu}^*(t) \geqslant \mu_* - \varepsilon$. Those hypothesis can be read as:*

$$\widehat{\mu}_{a_{t+1}}(t) \leqslant \mu_{a_{t+1}} + \varepsilon < \mu_* - \varepsilon \leqslant \mu^*(t). \tag{19}$$

*Lemma 4 and the monotonicity of the* KL *divergence implied by assumption 2, imply that:*

$$N_{a_{t+1}}(t) \leqslant \frac{\log t}{qd\left(\mu_{a_{t+1}} + \varepsilon \| \mu_* - \varepsilon\right)}. \tag{20}$$

*Property 2 then implies that there exists $\alpha_a$ such that*

$$N_{a_{t+1}}(t) \leqslant \frac{\log t}{qd\left(\mu_{a_{t+1}} \| \mu_*\right)} \left(1 + \alpha_a(\varepsilon)\right), \tag{21}$$

*with $\alpha_a(\varepsilon) \to 0$ as $\varepsilon \to 0$.*

*Proof.* Equation (19) is a direct consequence of the hypothesis of the Lemma. The strict inequality, $\mu_{a_{t+1}} < \mu_*$, implies that $a_{t+1}$ cannot belong to $\mathcal{A}_*(t)$. Hence, equation (16) from Lemma 4 applies and

$$qN_{a_{t+1}}(t)d\left(\widehat{\mu}_{a_{t+1}}(t) \| \widehat{\mu}^*\right) \leqslant \log t.$$

Because $\widehat{\mu}_{a_{t+1}}(t) < \mu_* - \varepsilon \leqslant \widehat{\mu}^*(t)$, using assumption 2 that implies $d\left(\widehat{\mu}_{a_{t+1}} \| \mu_* - \varepsilon\right) \leqslant d\left(\widehat{\mu}_{a_{t+1}} \| \widehat{\mu}^*(t)\right)$, we have that:

$$qN_{a_{t+1}}(t)d\left(\widehat{\mu}_{a_{t+1}} \| \mu_* - \varepsilon\right) \leqslant \log t.$$

Similarly, $\widehat{\mu}_{a_{t+1}}(t) \leqslant \mu_{a_{t+1}}(t) + \varepsilon < \mu_* - \varepsilon$ and using assumption 2 again, $d\left(\mu_a + \varepsilon \| \mu_* - \varepsilon\right) \leqslant d\left(\widehat{\mu}_{a_{t+1}} \| \mu_* - \varepsilon\right)$, we proved equation (20):

$$N_{a_{t+1}}(t) \leqslant \frac{\log t}{qd\left(\mu_{a_{t+1}} + \varepsilon \| \mu_* - \varepsilon\right)},$$

Using equation (11) from property 2, we deduce that there exists $\alpha_a$ as in property 2 such that:

$$N_{a_{t+1}}(t) \leqslant \frac{\log t}{qd\left(\mu_{a_{t+1}} \| \mu_*\right)} \left(1 + \alpha_a(\varepsilon)\right).$$

$\square$

In order to better clarify the proof of Theorem 2, we add two more lemmas that help emphasize where the hypothesis regarding the distribution space $\mathcal{F}$ are used. The first one is about decomposing the event of choosing a suboptimal arm.

**Intuition about Lemma 6** The intuition of the next lemma is the following. Let $a$ be a suboptimal arm. We will decompose the event of choosing arm $a$ at time $t + 1$, $\{a_{t+1} = a\}$, on three events. We recall that $\widehat{\mu}^*(t) = \max_{a \in \mathcal{A}} \widehat{\mu}_a(t)$. Let $\varepsilon$ be such that $0 < \varepsilon \leqslant \frac{1}{3} \min_{a \notin \mathcal{A}_*} (\mu_* - \mu_a)$ as in Lemma 5.

From Lemma 5, we know that under the event $\left\{\widehat{\mu}^*(t) \geqslant \mu_* - \varepsilon, \widehat{\mu}_{a_{t+1}} \leqslant \mu_{a_{t+1}} + \varepsilon\right\}$, the number $N_{a_{t+1}}$ is upper bounded by the desired asymptotic term. The intuition of this term is given in section 3 and made formal in Lemma 5. We want to play an arm as frequently as the likelihood of optimality of the group of size $q$ it may belong to.

Whatever the space of distributions $\mathcal{F}$, we will always assume a *concentration* around the mean property. Therefore, we know that the event $\left\{\widehat{\mu}_{a_{t+1}} > \mu_{a_{t+1}} + \varepsilon\right\}$ can be controlled by concentration

inequality ($a_{t+1}$ is the chosen arm at time $t$). The intuition for controlling this term is that one cannot play an arm too much while not having a good estimation of its expected reward.

The remainder of the two aforementioned events, $\{a_{t+1} \notin \mathcal{A}_*, \widehat{\mu}^*(t) < \mu_* - \varepsilon\}$, is the most technical and difficult to handle. An intuition may be the following. If $\widehat{\mu}^*(t) \leqslant \mu_* - \varepsilon$, then it is true that for any arm $k \in \mathcal{A}_*$, $\widehat{\mu}_k(t) \leqslant \mu_* - \varepsilon$. In that case, it means that with a *high enough* probability, most of the arm $k \in \mathcal{A}_*$ have not been pulled too much. The IMED and IMED-EC algorithms try to match likelihood of optimality with frequency of play. Therefore, we cannot keep *not* playing arm in $\mathcal{A}_*$ (*i.e.* we cannot play the suboptimal arm for *too* long before playing $k$) and when that happens, we will mostly observe a *regression toward the mean* making the event even more unlikely. We will rely on *concentration* tools to make the *regression toward the mean* statement more precise.

We now formalize those intuitions in Lemma 6.

**Lemma 6** (Fundamental decomposition). *Let $a$ be a suboptimal arm. Let $\varepsilon$ be such that $0 < \varepsilon \leqslant \frac{1}{3} \min_{a \notin \mathcal{A}_*} (\mu_* - \mu_a)$ as in Lemma 5. Under* IMED-EC, *shifting the time index $t$ by $|\mathcal{A}|$ (each arm is pulled once at the beginning), $N_a(t) = \sum_{t=1}^T \mathbb{1}\{a_{t+1} = a\}$, the number of time a suboptimal arm $a$ has been pulled after time $|\mathcal{A}|$ can be upper bounded by:*

$$\sum_{t=1}^T \mathbb{1}\{a_{t+1} = a\} \leqslant \sum_{t=1}^T \mathbb{1}\left\{a_{t+1} = a, N_a(t) \leqslant \frac{\log t}{qd(\mu_a + \varepsilon \| \mu_* - \varepsilon)}\right\} \tag{22}$$

$$+ \sum_{t=1}^T \mathbb{1}\{a_{t+1} = a, \widehat{\mu}^*(t) \geqslant \mu_* - \varepsilon, \widehat{\mu}_a(t) > \mu_a + \varepsilon\} \tag{23}$$

$$+ \sum_{t=1}^T \mathbb{1}\{a_{t+1} = a, \widehat{\mu}^*(t) < \mu_* - \varepsilon\}. \tag{24}$$

*Proof.* Following the aforementioned intuition, we decompose the event $\{a_{t+1} = a\}$ :

$$\begin{aligned}
\{a_{t+1} = a\} &= \{a_{t+1} = a, \widehat{\mu}^*(t) \geqslant \mu_* - \varepsilon\} \cup \{a_{t+1} = a, \widehat{\mu}^*(t) < \mu_* - \varepsilon\} \\
&= \{a_{t+1} = a, \widehat{\mu}^*(t) \geqslant \mu_* - \varepsilon, \widehat{\mu}_a(t) \leqslant \mu_a + \varepsilon\} \\
&\quad \cup \{a_{t+1} = a, \widehat{\mu}^*(t) \geqslant \mu_* - \varepsilon, \widehat{\mu}_a(t) > \mu_a + \varepsilon\} \\
&\quad \cup \{a_{t+1} = a, \widehat{\mu}^*(t) < \mu_* - \varepsilon\} \\
&= \{a_{t+1} = a, qN_a(t)d(\mu_a + \varepsilon \| \mu_* - \varepsilon) \leqslant \log t\} \qquad \text{\textit{Using Lemma 5}} \\
&\quad \cup \{a_{t+1} = a, \widehat{\mu}^*(t) \geqslant \mu_* - \varepsilon, \widehat{\mu}_a(t) > \mu_a + \varepsilon\} \\
&\quad \cup \{a_{t+1} = a, \widehat{\mu}^*(t) < \mu_* - \varepsilon\}.
\end{aligned}$$

Using indicators of those events and shifting the time index $t$ by $|\mathcal{A}|$ (each arm is pulled once at the beginning), we can upper bound $N_a(t) = \sum_{t=1}^T \mathbb{1}\{a_{t+1} = a\}$, the number of time a suboptimal arm $a$ as been pulled after time $|\mathcal{A}|$:

$$\sum_{t=1}^T \mathbb{1}\{a_{t+1} = a\} \leqslant \sum_{t=1}^T \mathbb{1}\left\{a_{t+1} = a, N_a(t) \leqslant \frac{\log t}{qd(\mu_a + \varepsilon \| \mu_* - \varepsilon)}\right\} \tag{22}$$

$$+ \sum_{t=1}^T \mathbb{1}\{a_{t+1} = a, \widehat{\mu}^*(t) \geqslant \mu_* - \varepsilon, \widehat{\mu}_a(t) > \mu_a + \varepsilon\} \tag{23}$$

$$+ \sum_{t=1}^T \mathbb{1}\{a_{t+1} = a, \widehat{\mu}^*(t) < \mu_* - \varepsilon\}. \tag{24}$$

$\square$

The second lemma is about bounding equation (24) by a quantity that can be controlled in both assumptions 2 and 3. This particular control is very specific to the $q$-equivalence structure and rely heavily on the fact that there is at least $q$ distributions in the optimal class $\mathcal{A}_*$.

**Lemma 7** (q-factorization). *Let $a$ be a suboptimal arm and $c \subseteq \mathcal{A}_*$ be any subset of $q$ optimal arms. Then, under the* IMED-EC *algorithm,*

$$\sum_{t=1}^{T} \mathbb{1}\left\{a_{t+1} = a, \widehat{\mu}^*(t) < \mu_* - \varepsilon\right\} \leqslant$$

$$\prod_{k \in c} \sum_{\substack{m_1 \geqslant 1 \\ \vdots \\ m_q \geqslant 1}} \mathbb{1}\left\{\widehat{\mu}_k^{m_k} < \mu_* - \varepsilon\right\} m_k \exp\left(m_k d\left(\widehat{\mu}_k^{m_k} \| \mu_* - \varepsilon\right)\right).$$

*Proof.* We want to control part (24) of the upper bound on the number of pulls of a suboptimal arm $a$. From Lemma 4, equation (15) and equation (17), we know that $\log\left(N_{a_{t+1}}(t)\right) \leqslant \min_{c_q \subseteq c_*} \sum_{k \in c_q} N_k(t) d\left(\widehat{\mu}_k(t) \| \widehat{\mu}^*\right) + \log N_k(t)$. Let $c \subseteq \mathcal{A}_*$ be any subset of $q$ optimal arms. Since we are studying the event $\{a_{t+1} = a\}$ this inequality becomes

$$\log\left(N_a(t)\right) \leqslant \min_{c_q \subseteq c_*} \sum_{k \in c_q} N_k(t) d\left(\widehat{\mu}_k(t) \| \widehat{\mu}^*\right) + \log N_k(t)$$

$$\leqslant \sum_{k \in c} N_k(t) d\left(\widehat{\mu}_k(t) \| \widehat{\mu}^*\right) + \log N_k(t).$$

We use this inequality to control the sum (24).

$$(24) = \sum_{t=1}^{T} \mathbb{1}\left\{a_{t+1} = a, \widehat{\mu}^*(t) < \mu_* - \varepsilon\right\}$$

$$= \sum_{t=1}^{T} \mathbb{1}\left\{\widehat{\mu}^*(t) < \mu_* - \varepsilon, \log N_a(t) \leqslant \sum_{k \in c} N_k(t) d\left(\widehat{\mu}_k(t) \| \widehat{\mu}^*(t)\right) + \log N_k(t)\right\} \times$$

$$\mathbb{1}\left\{a_{t+1} = a\right\}$$

$$= \sum_{t=1}^{T} \sum_{n=1}^{T} \mathbb{1}\left\{\widehat{\mu}^*(t) < \mu_* - \varepsilon, \log(n) \leqslant \sum_{k \in c} N_k(t) d\left(\widehat{\mu}_k(t) \| \widehat{\mu}^*(t)\right) + \log N_k(t)\right\} \times$$

$$\mathbb{1}\left\{a_{t+1} = a, N_a(t) = n\right\}.$$

Since $\widehat{\mu}^*(t) < \mu_* - \varepsilon$ and $\widehat{\mu}^*(t) = \max_{b \in \mathcal{A}} \widehat{\mu}_b(t)$, we can use the monotonicity of the KL divergence to state that for all $k \in c$, $d\left(\widehat{\mu}_k^{m_k} \| \widehat{\mu}^*(t)\right) \leqslant d\left(\widehat{\mu}_k(t) \| \mu_* - \varepsilon\right)$. This implies the inclusion of events,

$$\left\{\widehat{\mu}^*(t) < \mu_* - \varepsilon, \log(n) \leqslant \sum_{k \in c} N_k(t) d\left(\widehat{\mu}_k(t) \| \widehat{\mu}^*(t)\right) + \log N_k(t)\right\} \subseteq$$

$$\left\{\widehat{\mu}^*(t) < \mu_* - \varepsilon, \log(n) \leqslant \sum_{k \in c} N_k(t) d\left(\widehat{\mu}_k(t) \| \mu_* - \varepsilon\right) + \log N_k(t)\right\},$$

which can be used to control the indicators. Furthermore, $\widehat{\mu}^*(t) \leqslant \mu_* - \varepsilon$ implies that $\max_{k \in c} \widehat{\mu}_k(t) \leqslant \mu_* - \varepsilon$ since $\max_{k \in c} \widehat{\mu}_k(t) \leqslant \max_{k \in \mathcal{A}} \widehat{\mu}_k(t)$. Therefore,

$$\left\{\widehat{\mu}^*(t) < \mu_* - \varepsilon, \log(n) \leqslant \sum_{k \in c} N_k(t) d\left(\widehat{\mu}_k(t) \| \widehat{\mu}^*(t)\right) + \log N_k(t)\right\} \subseteq$$

$$\left\{\max_{k \in c} \widehat{\mu}_k(t) < \mu_* - \varepsilon, \log(n) \leqslant \sum_{k \in c} N_k(t) d\left(\widehat{\mu}_k(t) \| \mu_* - \varepsilon\right) + \log N_k(t)\right\},$$

which we use to control the indicators. We then obtain

$$(24) \leqslant \sum_{t=1}^{T} \sum_{n=1}^{T} \mathbb{1} \left\{ \max_{k \in c} \widehat{\mu}_k^{m_k} < \mu_* - \varepsilon, \log(n) \leqslant \sum_{k \in c} N_k(t) d\left(\widehat{\mu}_k(t) \| \mu_* - \varepsilon\right) + \log N_k(t) \right\} \times$$

$$\mathbb{1}\{a_{t+1} = a, N_a(t) = n\}$$

$$= \sum_{\substack{m_1 \geqslant 1 \\ \vdots \\ m_q \geqslant 1}} \sum_{n \geqslant 1} \sum_{t=1}^{T} \mathbb{1} \left\{ \max_{k \in c} \widehat{\mu}_k^{m_k} < \mu_* - \varepsilon, \log(n) \leqslant \sum_{k \in c} m_k d\left(\widehat{\mu}_k^{m_k} \| \mu_* - \varepsilon\right) + \log m_k \right\} \times$$

$$\mathbb{1}\{a_{t+1} = a, N_a(t) = n\} \underbrace{\prod_{k \in c} \mathbb{1}\{N_k(t) = m_k\}}_{\leqslant 1}$$

$$\leqslant \sum_{\substack{m_1 \geqslant 1 \\ \vdots \\ m_q \geqslant 1}} \sum_{n \geqslant 1} \sum_{t=1}^{T} \mathbb{1} \left\{ \max_{k \in c} \widehat{\mu}_k^{m_k} < \mu_* - \varepsilon, \log(n) \leqslant \sum_{k \in c} m_k d\left(\widehat{\mu}_k^{m_k} \| \mu_* - \varepsilon\right) + \log m_k \right\} \times$$

$$\mathbb{1}\{a_{t+1} = a, N_a(t) = n\}$$

$$= \sum_{\substack{m_1 \geqslant 1 \\ \vdots \\ m_q \geqslant 1}} \sum_{n \geqslant 1} \mathbb{1} \left\{ \max_{k \in c} \widehat{\mu}_k^{m_k} < \mu_* - \varepsilon, \log(n) \leqslant \sum_{k \in c} m_k d\left(\widehat{\mu}_k^{m_k} \| \mu_* - \varepsilon\right) + \log m_k \right\} \times$$

$$\underbrace{\sum_{t=1}^{T} \mathbb{1}\{a_{t+1} = a, N_a(t) = n\}}_{\leqslant 1}$$

$$\leqslant \sum_{\substack{m_1 \geqslant 1 \\ \vdots \\ m_q \geqslant 1}} \sum_{n \geqslant 1} \mathbb{1} \left\{ \max_{k \in c} \widehat{\mu}_k^{m_k} < \mu_* - \varepsilon, \log(n) \leqslant \sum_{k \in c} m_k d\left(\widehat{\mu}_k^{m_k} \| \mu_* - \varepsilon\right) + \log m_k \right\}.$$

We can then factorize the following term

$$\mathbb{1} \left\{ \max_{k \in c} \widehat{\mu}_k^{m_k} < \mu_* - \varepsilon, \log(n) \leqslant \sum_{k \in c} m_k d\left(\widehat{\mu}_k^{m_k} \| \mu_* - \varepsilon\right) + \log m_k \right\} =$$

$$\mathbb{1} \left\{ \max_{k \in c} \widehat{\mu}_k^{m_k} < \mu_* - \varepsilon \right\} \mathbb{1} \left\{ \log(n) \leqslant \sum_{k \in c} m_k d\left(\widehat{\mu}_k^{m_k} \| \mu_* - \varepsilon\right) + \log m_k \right\},$$

and remark that $\mathbb{1}\left\{\max_{k \in c} \widehat{\mu}_k^{m_k} < \mu_* - \varepsilon\right\}$ does not depend on $n$. Hence,

$$(24) \leqslant \sum_{\substack{m_1 \geqslant 1 \\ \vdots \\ m_q \geqslant 1}} \sum_{n \geqslant 1} \mathbb{1}\left\{\max_{k \in c} \widehat{\mu}_k^{m_k} < \mu_* - \varepsilon, \log(n) \leqslant \sum_{k \in c} m_k d\left(\widehat{\mu}_k^{m_k} \| \mu_* - \varepsilon\right) + \log m_k\right\}$$

$$= \sum_{\substack{m_1 \geqslant 1 \\ \vdots \\ m_q \geqslant 1}} \mathbb{1}\left\{\max_{k \in c} \widehat{\mu}_k^{m_k} < \mu_* - \varepsilon\right\} \sum_{n \geqslant 1} \mathbb{1}\left\{\log(n) \leqslant \sum_{k \in c} m_k d\left(\widehat{\mu}_k^{m_k} \| \mu_* - \varepsilon\right) + \log m_k\right\}$$

$$\leqslant \sum_{\substack{m_1 \geqslant 1 \\ \vdots \\ m_q \geqslant 1}} \mathbb{1}\left\{\max_{k \in c} \widehat{\mu}_k^{m_k} < \mu_* - \varepsilon\right\} \exp\left(\sum_{k \in c} m_k d\left(\widehat{\mu}_k^{m_k} \| \mu_* - \varepsilon\right) + \log m_k\right).$$

Since $\exp\left(\sum_{k \in c} m_k d\left(\widehat{\mu}_k^{m_k} \| \mu_* - \varepsilon\right) + \log m_k\right) = \prod_{k \in c} m_k \exp\left(m_k d\left(\widehat{\mu}_k^{m_k} \| \mu_* - \varepsilon\right)\right)$ and $\mathbb{1}\left\{\max_{k \in c} \widehat{\mu}_k^{m_k} < \mu_* - \varepsilon\right\} = \prod_{k \in c} \mathbb{1}\left\{\widehat{\mu}_k^{m_k} < \mu_* - \varepsilon\right\}$, we can rewrite the last bound as

$$(24) \leqslant \sum_{\substack{m_1 \geqslant 1 \\ \vdots \\ m_q \geqslant 1}} \prod_{k \in c} \mathbb{1}\left\{\widehat{\mu}_k^{m_k} < \mu_* - \varepsilon\right\} m_k \exp\left(m_k d\left(\widehat{\mu}_k^{m_k} \| \mu_* - \varepsilon\right)\right)$$

$$\leqslant \prod_{k \in c} \sum_{\substack{m_1 \geqslant 1 \\ \vdots \\ m_q \geqslant 1}} \mathbb{1}\left\{\widehat{\mu}_k^{m_k} < \mu_* - \varepsilon\right\} m_k \exp\left(m_k d\left(\widehat{\mu}_k^{m_k} \| \mu_* - \varepsilon\right)\right).$$

where we used the fact that a sum of product of non-negative terms is not greater than the product of the sum of these terms, since one contains all the terms of the other. $\qquad\square$

Thanks to Lemma 7, controlling the Equation (24) amounts to controlling terms like

$$\mathbb{E}\left(\sum_{m_k \geqslant 1} \mathbb{1}\left\{\widehat{\mu}_k^{m_k} < \mu_* - \varepsilon\right\} m_k \exp\left(m_k d\left(\widehat{\mu}_k^{m_k} \| \mu_* - \varepsilon\right)\right)\right), \qquad (25)$$

which is linked to the upper bound one can have on $\mathbb{P}\left(d\left(\widehat{\mu}_k^{m_k} \| \mu_* - \varepsilon\right) \geqslant u, \widehat{\mu}_k^{m_k} \leqslant \mu_* - \varepsilon\right)$ and whether or not it is better than $\exp(-m_k u)$. As we can see, controlling this terms amounts to a *concentration* hypothesis of the distributions in $\mathcal{F}$. Assumption 2 and assumption 3 are two possible assumptions that make it possible to control the Equation (25).

We are now ready to prove the Theorem 2.

*Proof of Theorem 2.* Let $a$ be a suboptimal arm and let $\varepsilon$ be such that $0 < \varepsilon \leqslant \frac{1}{3} \min_{a \notin \mathcal{A}_*} (\mu_* - \mu_a)$. By Lemma 6 we have the following decomposition:

$$\sum_{t=1}^{T} \mathbb{1}\{a_{t+1} = a\} \leqslant \sum_{t=1}^{T} \mathbb{1}\left\{a_{t+1} = a, N_a(t) \leqslant \frac{\log t}{qd(\mu_a + \varepsilon \| \mu_* - \varepsilon)}\right\} \qquad (22)$$

$$+ \sum_{t=1}^{T} \mathbb{1}\left\{a_{t+1} = a, \widehat{\mu}^*(t) \geqslant \mu_* - \varepsilon, \widehat{\mu}_a(t) > \mu_a + \varepsilon\right\} \qquad (23)$$

$$+ \sum_{t=1}^{T} \mathbb{1}\left\{a_{t+1} = a, \widehat{\mu}^*(t) < \mu_* - \varepsilon\right\}. \qquad (24)$$

**Control of equation (22)**  Equation (22) can be controlled as a random variable without any concentration tools. The following derivation bound the sum (22) by:

$$
(22) = \sum_{t=1}^{T} \mathbb{1}\left\{a_{t+1} = a, N_a(t) \leqslant \frac{\log t}{qd\left(\mu_a + \varepsilon \| \mu_* - \varepsilon\right)}\right\}
$$

$$
= \sum_{n=1}^{T} \sum_{t=1}^{T} \mathbb{1}\left\{n \leqslant \frac{\log t}{qd\left(\mu_a + \varepsilon \| \mu_* - \varepsilon\right)}\right\} \mathbb{1}\left\{a_{t+1} = a, N_a(t) = n\right\}
$$

$$
= \sum_{n=1}^{T} \mathbb{1}\left\{n \leqslant \frac{\log T}{qd\left(\mu_a + \varepsilon \| \mu_* - \varepsilon\right)}\right\} \underbrace{\sum_{t=1}^{T} \mathbb{1}\left\{a_{t+1} = a, N_a(t) = n\right\}}_{\leqslant 1}
$$

$$
\leqslant \sum_{n=1}^{T} \mathbb{1}\left\{n \leqslant \frac{\log T}{qd\left(\mu_a + \varepsilon \| \mu_* - \varepsilon\right)}\right\}
$$

$$
= \left\lfloor \frac{\log T}{qd\left(\mu_a + \varepsilon \| \mu_* - \varepsilon\right)} \right\rfloor
$$

$$
\leqslant \frac{\log T}{qd\left(\mu_a + \varepsilon \| \mu_* - \varepsilon\right)}.
$$

The derivation relies on the simple fact that an indicator function is upper bounded by 1. This part proved that:

$$
\sum_{t=1}^{T} \mathbb{1}\left\{a_{t+1} = a\right\} \leqslant \frac{\log T}{qd\left(\mu_a + \varepsilon \| \mu_* - \varepsilon\right)} \tag{22'}
$$

$$
+ \sum_{t=1}^{T} \mathbb{1}\left\{a_{t+1} = a, \widehat{\mu}^*(t) \geqslant \mu_* - \varepsilon, \widehat{\mu}_a(t) > \mu_a + \varepsilon\right\} \tag{23}
$$

$$
+ \sum_{t=1}^{T} \mathbb{1}\left\{a_{t+1} = a, \widehat{\mu}^*(t) < \mu_* - \varepsilon\right\}. \tag{24}
$$

**Control of equation (23)**  Equation (23) can be controlled using large deviation hypothesis on the set of distributions that are considered. It should be noted that this term is also bounded by $O(1)$ in the paper Honda and Takemura [2015]. Therefore, this term can be also be handled under assumption 3. We give an upper bound under assumption 2. The common fact of those two assumptions is the *light-tail* property of the considered distributions. A distributions is said *light-tailed* if its moment generating function exists in a neighborhood of $0$. In that case one can apply a concentration property

since Cramer's theorem applies (see [Dembo and Zeitouni, 1998, Theorem 2.2.3]).

$$(23) = \sum_{t=1}^{T} \mathbb{1}\left\{a_{t+1} = a, \widehat{\mu}^*(t) \geqslant \mu_* - \varepsilon, \widehat{\mu}_a(t) > \mu_a + \varepsilon\right\}$$

$$\leqslant \sum_{t=1}^{T} \mathbb{1}\left\{a_{t+1} = a, \widehat{\mu}_a(t) > \mu_a + \varepsilon\right\}$$

$$= \sum_{n=1}^{T} \sum_{t=1}^{T} \mathbb{1}\left\{a_{t+1} = a, N_a(t) = n, \widehat{\mu}_a(t) > \mu_a + \varepsilon\right\}$$

$$= \sum_{n=1}^{T} \sum_{t=1}^{T} \mathbb{1}\left\{a_{t+1} = a, N_a(t) = n\right\} \mathbb{1}\left\{\widehat{\mu}_a^n > \mu_a + \varepsilon\right\}$$

$$= \sum_{n=1}^{T} \mathbb{1}\left\{\widehat{\mu}_a^n > \mu_a + \varepsilon\right\} \underbrace{\sum_{t=1}^{T} \mathbb{1}\left\{a_{t+1} = a, N_a(t) = n\right\}}_{\leqslant 1}$$

$$\leqslant \sum_{n=1}^{T} \mathbb{1}\left\{\widehat{\mu}_a^n > \mu_a + \varepsilon\right\}.$$

Taking the expectation of both sides,

$$\mathbb{E}\left((23)\right) \leqslant \sum_{n=1}^{T} \mathbb{P}\left(\widehat{\mu}_a^n > \mu_a + \varepsilon\right),$$

which is a series of positive real numbers between $0$ and $1$ if we set $T = +\infty$. The reason we are interested in the limit is because we want $\mathbb{E}\left((23)\right)$ to be upper bounded by a time independent quantity. For this series to be convergent, we need the terms of the series, $\left(\mathbb{P}\left(\widehat{\mu}_a(n) > \mu_a + \varepsilon\right)\right)_n$ to converge fast enough toward $0$. Denoting $\varphi_a$ the moment generating function of arm $a$ and $\psi_a = \log \varphi_a$ the cumulant generating function we derive that for all $\lambda > 0$:

$$\mathbb{P}\left(\widehat{\mu}_a^n > \mu_a + \varepsilon\right) = \mathbb{P}\left(\sum_{i=1}^{n} X_i^a > n\left(\mu_a + \varepsilon\right)\right)$$

$$\leqslant \frac{\varphi_a\left(\lambda\right)^n}{e^{n\lambda(\mu_a + \varepsilon)}} \qquad \text{Markov inequality}$$

$$= \exp\left(n\left(\psi_a\left(\lambda\right) - \lambda\left(\mu_a + \varepsilon\right)\right)\right)$$

$$= \exp\left(-n\left(\lambda\left(\mu_a + \varepsilon\right) - \psi_a\left(\lambda\right)\right)\right).$$

Since this inequality is true for all $\lambda$ we can minimize the right-hand side expression which features the Legendre-Fenchel transformation of $\psi_a$ (also known as the Cramer transform). We denote $\psi_a^*(\varepsilon) = \sup_\lambda \left(\lambda\left(\mu_a + \varepsilon\right) - \psi_a\left(\lambda\right)\right)$ the Legendre-Fenchel transform of $\psi_a$ that exists thanks to assumption 2.

For distributions in $\mathcal{F}$ this quantity is strictly positive as it is proved in Dembo and Zeitouni [1998]. Therefore, we proved that

$$\mathbb{P}\left(\widehat{\mu}_a^n > \mu_a + \varepsilon\right) \leqslant \exp\left(-n\psi_a^*\left(\varepsilon\right)\right)$$

$$= \left(\exp\left(-\psi_a^*\left(\varepsilon\right)\right)\right)^n,$$

with $0 \leqslant \exp\left(-\psi_a^*(\varepsilon)\right) < 1$. This result is enough to bound equation (23) in expectation,

$$
\begin{aligned}
\mathbb{E}\left((23)\right) &\leqslant \sum_{n=1}^{T} \mathbb{P}\left(\widehat{\mu}_a^n > \mu_a + \varepsilon\right) \\
&\leqslant \sum_{n=1}^{+\infty} \mathbb{P}\left(\widehat{\mu}_a^n > \mu_a + \varepsilon\right) \\
&\leqslant \sum_{n=1}^{+\infty} \exp\left(-n\psi_a^*(\varepsilon)\right) \\
&= C_a(\varepsilon),
\end{aligned}
$$

where $C_a(\varepsilon)$ denotes the limit of the series $\sum_{n=1}^{+\infty} \exp\left(-n\psi_a^*(\varepsilon)\right)$. This part and the previous one proved that

$$
\mathbb{E}\left(\sum_{t=1}^{T} \mathbb{1}\left\{a_{t+1} = a\right\}\right) \leqslant \frac{\log T}{qd\left(\mu_a + \varepsilon \| \mu_* - \varepsilon\right)} \tag{22'}
$$

$$
+ C_a(\varepsilon) \tag{23'}
$$

$$
+ \mathbb{E}\left(\sum_{t=1}^{T} \mathbb{1}\left\{a_{t+1} = a, \widehat{\mu}^*(t) < \mu_* - \varepsilon\right\}\right). \tag{24}
$$

**Control of equation (24)**   We are left to control part (24) of the upper bound on the number of pulls of a suboptimal arm. Let $c \subseteq \mathcal{A}_*$ be any subset of $q$ optimal arms. From the q-factorization lemma, Lemma 7, we know that

$$
\sum_{t=1}^{T} \mathbb{1}\left\{a_{t+1} = a, \widehat{\mu}^*(t) < \mu_* - \varepsilon\right\} \leqslant
$$

$$
\prod_{k \in c} \sum_{\substack{m_1 \geqslant 1 \\ \vdots \\ m_q \geqslant 1}} \mathbb{1}\left\{\widehat{\mu}_k^{m_k} < \mu_* - \varepsilon\right\} m_k \exp\left(m_k d\left(\widehat{\mu}_k^{m_k} \| \mu_* - \varepsilon\right)\right).
$$

Since samples from the different arms are independents, the inequality holds in expectation:

$$
\mathbb{E}\left((24)\right) \leqslant \prod_{k \in c} \mathbb{E}\left(\sum_{\substack{m_1 \geqslant 1 \\ \vdots \\ m_q \geqslant 1}} \mathbb{1}\left\{\widehat{\mu}_k^{m_k} < \mu_* - \varepsilon\right\} m_k \exp\left(m_k d\left(\widehat{\mu}_k^{m_k} \| \mu_* - \varepsilon\right)\right)\right).
$$

The proof of Lemma 14 in Honda and Takemura [2015], from equation (26), features a very similar quantity. In particular, it has been proved that for all $k \in c$, under the assumption 3

$$
\mathbb{E}\left(\sum_{m_k \geqslant 1} \mathbb{1}\left\{\widehat{\mu}_k^{m_k} < \mu_* - \varepsilon\right\} m_k \exp\left(m_k d\left(\widehat{\mu}_k^{m_k} \| \mu_* - \varepsilon\right)\right)\right) \leqslant D_{\textit{IMED}}(\varepsilon), \tag{26}
$$

where $D_{\textit{IMED}}(\varepsilon)$ is given by equation (28) of Honda and Takemura [2015]. The proof of Lemma 14 in Honda and Takemura [2015] relies on the Proposition 11 of Honda and Takemura [2015]. Under assumption 2, Proposition 1 allows us to upper bound the left hand side of Equation (27) by a time independent quantity using the same strategy of integration by parts used in Honda and Takemura [2015]:

$$
\mathbb{E}\left(\sum_{m_k \geqslant 1} \mathbb{1}\left\{\widehat{\mu}_k^{m_k} < \mu_* - \varepsilon\right\} m_k \exp\left(m_k d\left(\widehat{\mu}_k^{m_k} \| \mu_* - \varepsilon\right)\right)\right) \leqslant D(\varepsilon). \tag{27}
$$

This upper bounds amount to the fact that $u \mapsto \exp\left(-\alpha\sqrt{u}\right)$ is integrable on $[0, +\infty[$ for all $\alpha > 0$. Let $P(u) = \mathbb{P}_{\nu_k}\left(d\left(\widehat{\mu}_k^{m_k}\|\mu_* - \varepsilon\right) > u, \widehat{\mu}_k^{\mu_k} < \mu_* - \varepsilon\right)$. Under assumption 2, Proposition 1 upper bounds $P(u)$, $P(u) \leqslant \exp\left(-m_k\left(u + \lambda\left(\mu_*, \mu_* - \varepsilon\right)\sqrt{u}\right)\right)$ with $\lambda\left(\mu_*, \mu_* - \varepsilon\right) > 0$. Therefore, we can integrate against $P(u)$

$$\sum_{m_k \geqslant 1} \mathbb{E}\left(\mathbb{1}\left\{\widehat{\mu}_k^{m_k} < \mu_* - \varepsilon\right\} m_k \exp\left(m_k d\left(\widehat{\mu}_k^{m_k}\|\mu_* - \varepsilon\right)\right)\right)$$

$$= \int_0^\infty m_k \exp\left(m_k u\right)\left(-dP(u)\right)$$

$$= \sum_{m_k \geqslant 1}\left[m_k \exp\left(m_k u\right)\left(-P(u)\right)\right]_0^\infty + \int_0^\infty m_k^2 \exp\left(m_k u\right) P(u) du$$

$$\leqslant h\left(\varepsilon\right) + \sum_{m_k \geqslant 1}\int_0^\infty m_k^2 \exp\left(m_k u\right) \exp\left(-m_k\left(u + \lambda\left(\mu_*, \mu_* - \varepsilon\right)\sqrt{u}\right)\right) du$$

$$= h\left(\varepsilon\right) + \sum_{m_k \geqslant 1}\int_0^\infty m_k^2 \exp\left(-m_k\lambda\left(\mu_*, \mu_* - \varepsilon\right)\sqrt{u}\right) du$$

$$\leqslant h\left(\varepsilon\right) + p\left(\varepsilon\right)$$

$$= D\left(\varepsilon\right),$$

which proves our claim. Therefore, under assumption 2 or assumption 3, we can deduce a bound on equation (24),

$$(24) \leqslant \prod_{k \in c}\sum_{\substack{m_1 \geqslant 1 \\ \vdots \\ m_q \geqslant 1}} \mathbb{1}\left\{\widehat{\mu}_k^{m_k} < \mu_* - \varepsilon\right\} m_k \exp\left(m_k d\left(\widehat{\mu}_k^{m_k}\|\mu_* - \varepsilon\right)\right)$$

$$\leqslant D\left(\varepsilon\right)^q,$$

because $c$ is a set of size $q$.

All combined, those derivations proved that

$$\mathbb{E}\left(\sum_{t=1}^T \mathbb{1}\left\{a_{t+1} = a\right\}\right) \leqslant \frac{\log T}{qd\left(\mu_a + \varepsilon\|\mu_* - \varepsilon\right)} \qquad (22')$$

$$+ C_a\left(\varepsilon\right) \qquad (23')$$

$$+ D(\varepsilon)^q, \qquad (24')$$

and can be written as

$$\mathbb{E}\left(N_a(t)\right) \leqslant \frac{\log T}{qd\left(\mu_a + \varepsilon\|\mu_* - \varepsilon\right)} + C_a\left(\varepsilon\right) + D(\varepsilon)^q.$$

$\square$

Using property 2 for the KL divergence, we finally get the final expression proving the Theorem 2,

$$\mathbb{E}\left(N_a(t)\right) \leqslant \frac{\log T}{qd\left(\mu_a\|\mu_*\right)}\left(1 + \alpha_a(\varepsilon)\right) + f(\varepsilon),$$

with $f(\varepsilon) = C_a\left(\varepsilon\right) + D(\varepsilon)^q$.

*Proof of Theorem 3.* The Theorem 2 we just proved states that

$$\mathbb{E}\left(N_a(t)\right) \leqslant \frac{\log T}{qd\left(\mu_a\|\mu_*\right)}\left(1 + \alpha_a(\varepsilon)\right) + f(\varepsilon)$$

with $f(\varepsilon) = C_a\left(\varepsilon\right) + D(\varepsilon)^q$. Dividing both sides by $\log T$, we get that for all $\varepsilon$ small enough

$$\liminf_{t \to +\infty}\frac{\mathbb{E}_\nu\left(N_a(T)\right)}{\log T} \leqslant \frac{1}{qd\left(\mu_a\|\mu_*\right)}\left(1 + \alpha_a(\varepsilon)\right).$$

Letting $\varepsilon$ tends to 0 proves the Theorem 3. $\square$

## D   Experiments

In this section, we illustrate the performances of `IMED-EC` with a few more plots and explore the *dispatching* of this algorithm. By *dispatching*, we mean to compare the discrepancy in the number of pulls within a class. In particular, we are interested in the behavior of the different sampling strategies within the optimal class. The different settings are the same as the one presented in the main part of this paper. We present additional details.

**Balanced class, perfect knowledge**   In this set of experiments, see Figure 5, we focus on the bandit configurations in which all equivalence classes have the same cardinality and assume that we know the number of elements per class. Recall that since `OSSB` has to solve a combinatorial optimization problem at each time step, we can cannot carry experiments on large set of arms while comparing `IMED-EC` to it. In this particular setting, we see that while `OSSB` and `IMED-EC` are provably

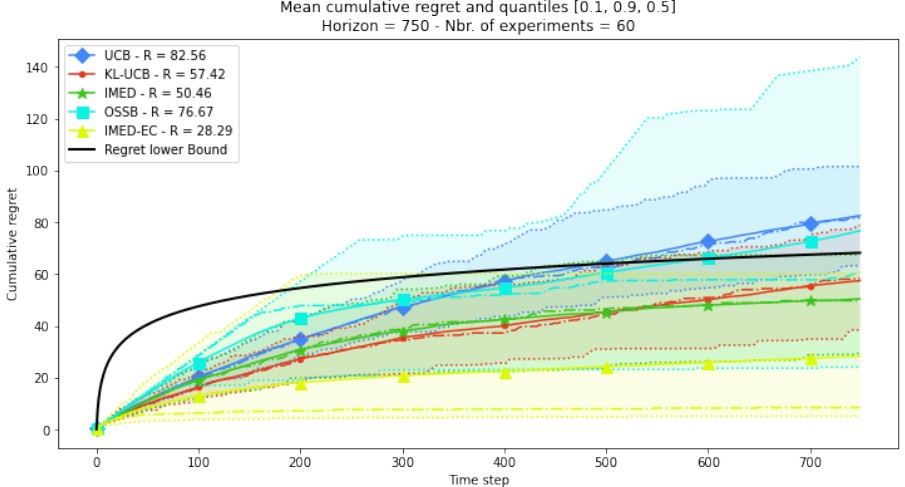

Figure 5: 3 classes, 2 distributions per class - set of means = $\{0.1, 0.3, 0.7\}$

asymptotically optimal, `IMED-EC` numerically performs better in finite time horizon.

Next, we explore the *dispatching* of `IMED-EC` and compare it to the one of `KLUCB` and `IMED`. We run the three algorithms 1000 times on an bandit problem whose number of class is 4 with means $\{0.1, 0.3, 0.6, 0.9\}$ and 10 Gaussian distributions with unit variance within each class. The chosen horizon is 2000. We assume that `IMED-EC` has perfect knowledge on the number of elements per class, 10. In the next section, similar plots (see Figure 11, Figure 12, Figure 13 and Figure 14) can be found where `IMED-EC` only knows a strict lower bound on the number of elements per class. For each of the 4 classes, we report the histogram accounting for the number of times distributions within each class has been pulled. Specifically, we are interested in the statistical order of the number of pulls within each class. After each run, for each class, we sort the number of pulls. The histograms are built using those sorted number of pulls. Error bars corresponds to the standard deviations and have been clipped to not go below the $x$-axis.

For the most suboptimal class, Figure 6, not much can be said since the number of pulls is very low. Still, one can see that the progression of the order statistics for `KLUCB` and `IMED` is somewhat *linear* while it seems more *exponential* for `IMED-EC`. (Note that we use these terms here informally.)

The same linear versus exponential apparent behavior can be seen on Figure 7.

On Figure 8, one can clearly a difference in the behaviour of `KLUCB` and `IMED`, that have *small* error bars, and `IMED-EC` that have a large error bar for the most pulled arm within the least suboptimal class. We can clearly see how risky it might be to reduce the exploration from this error bar. However, this risk is compensated by the fact that there is at least $q$ similar distributions. This can be read from the fact that the sum of all the number of pulls within this class for `KLUCB` and `IMED` is above 200 which is roughly the maximal number of pulls of `IMED-EC` within this class computed using the upper bounds (given by the maximal value of the standard deviation).

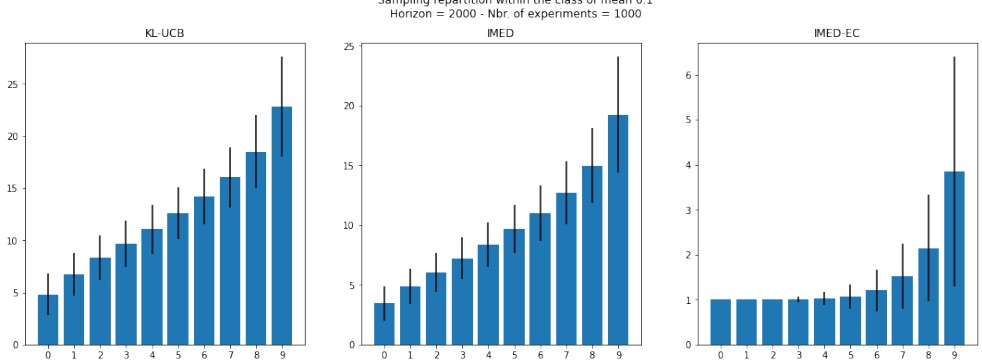

Figure 6: 4 classes, 10 distributions per class - set of means = $\{\mathbf{0.1}, 0.3, 0.6, 0.9\}$ - class of mean 0.1

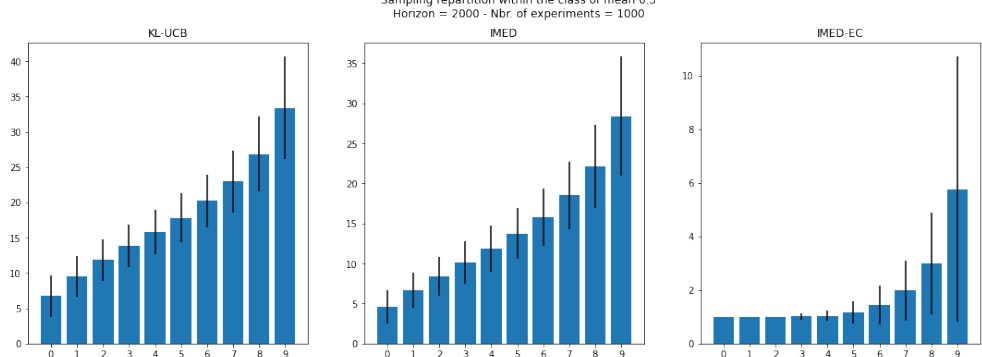

Figure 7: 4 classes, 10 distributions per class - set of means = $\{0.1, \mathbf{0.3}, 0.6, 0.9\}$ - class of mean 0.3

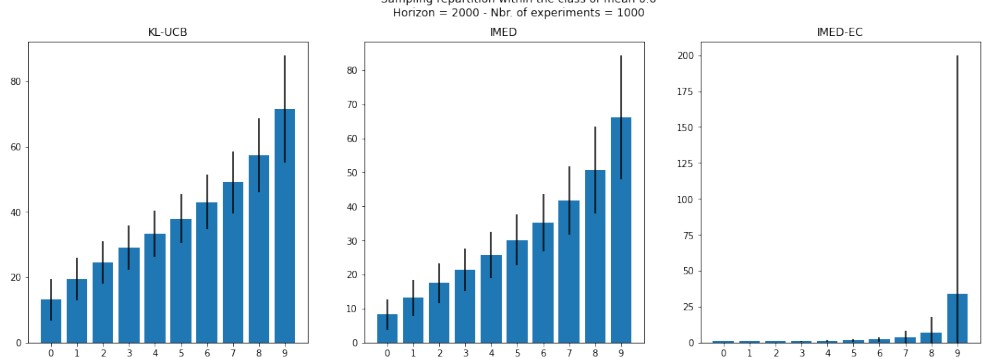

Figure 8: 4 classes, 10 distributions per class - set of means = $\{0.1, 0.3, \mathbf{0.6}, 0.9\}$ - class of mean 0.6

Finally, Figure 9 enables to compare the behaviours of the algorithms within the optimal class. It seems clear that, at least numerically, IMED-EC is not a fair algorithm in finite time (in the sense that it does not equally distribute the pulls between arms from the same class) and that it leverages the lower bound on the number of elements per class to play a more risky strategy, and benefits from it. Again, we observe the linear versus exponential progression in the order statistics of the number of pulls.

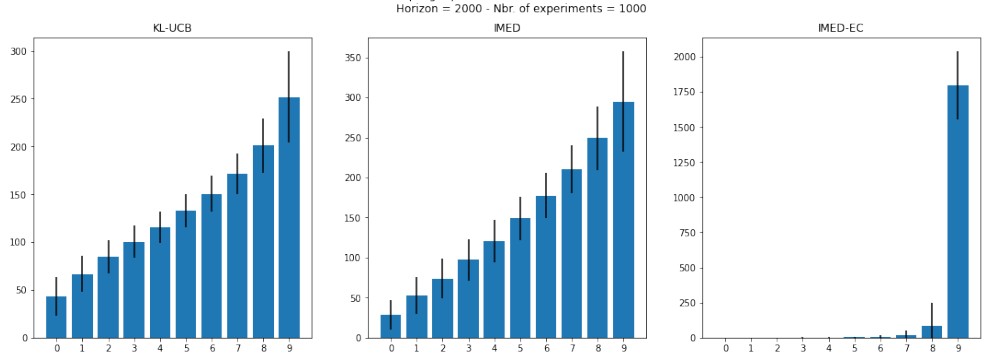

Figure 9: 4 classes, 10 distributions per class - set of means = $\{0.1, 0.3, 0.6, \mathbf{0.9}\}$ - class of mean $0.9$

**Imperfect knowledge**   In the experiment plotted Figure 10, we leverage the knowledge hypothesis and assume that we only know a lower bound on the number of elements per class while the classes are still balanced. We compare `IMED-EC` to unspecialized bandit algorithm, `IMED` and `KLUCB`. We can

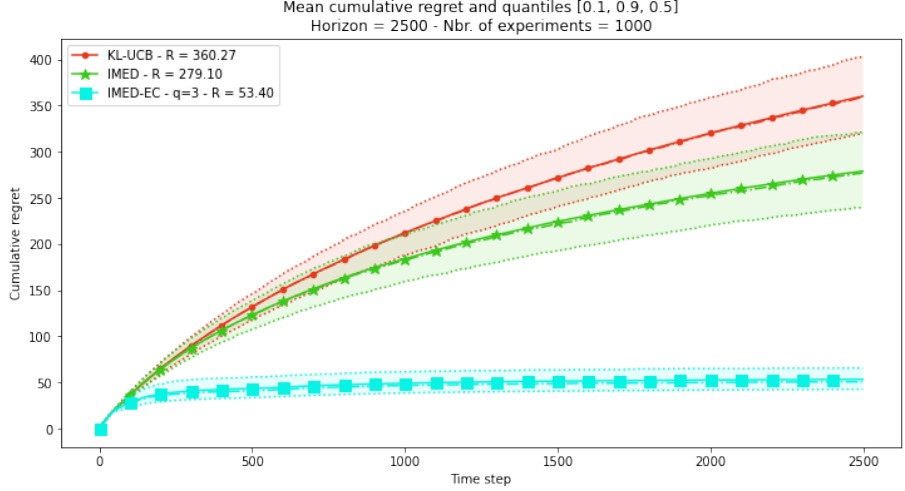

Figure 10: 7 classes, 10 distributions per class - set of means = $\{0.1, 0.3, 0.4, 0.5, 0.6, 0.75, 0.9\}$

see that the finite time cumulative regret of `IMED-EC` indeed is much smaller than the regret of the unspecialized algorithms, showing the ability of `IMED-EC` to effectively exploit this weak knowledge.

For the sake of completeness, we explore the *dispatching* of `IMED-EC` and compare it to the one of `KLUCB` and `IMED` on this setting. We run the three algorithms $1000$ times on an bandit problem whose number of class is $4$ with means $\{0.1, 0.3, 0.6, 0.9\}$ and 10 Gaussian distributions with unit variance within each class. The chosen horizon is $2000$. We assume that `IMED-EC` does not have perfect knowledge on the number of elements per class, and we use 3 as the lower bound parameter of `IMED-EC`. For each of the $4$ classes, we report the histogram accounting for the number of times distributions within each class has been pulled.Error bars corresponds to the standard deviations and have been clipped to not go below the $x$-axis.

Comments that were respectively made for Figure 6, Figure 7, Figure 8, and Figure 9 can similarly made for Figure 11, Figure 12, Figure 13, and Figure 14.

Interestingly, we can tell apart the two settings by looking at the behaviour of the algorithms for the least suboptimal class, *i.e.* comparing Figure 8 and Figure 13. In Figure 8 the error bar for the most pulled elements is much larger than in Figure 13 meaning that the algorithm at stakes explore less.

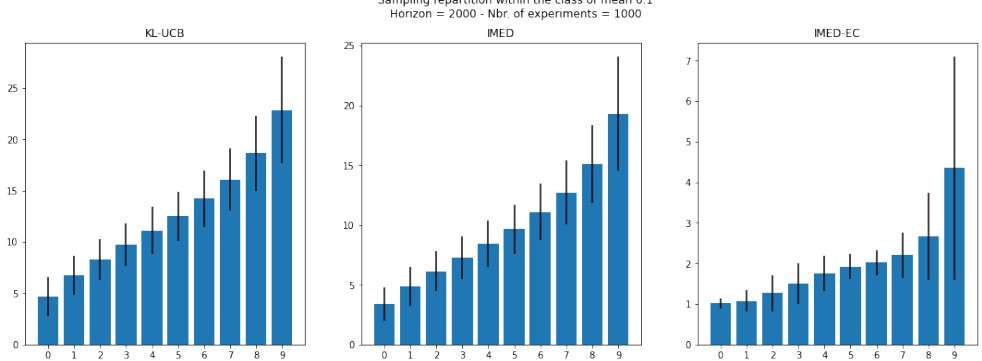

Figure 11: 4 classes, 10 distributions per class - set of means = $\{\mathbf{0.1}, 0.3, 0.6, 0.9\}$ - mean 0.1

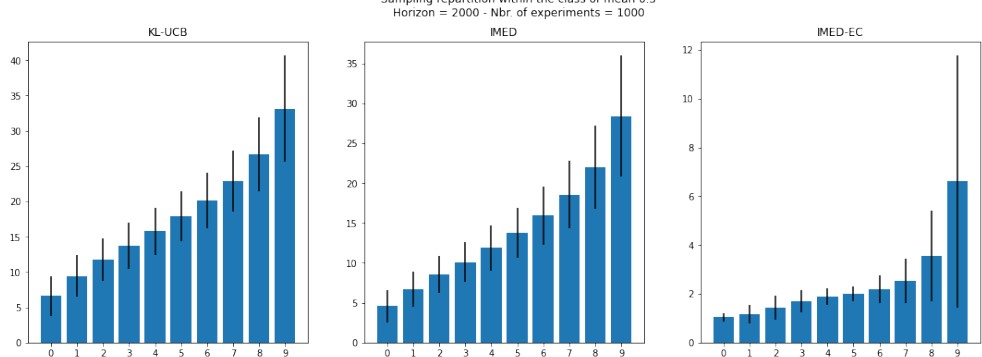

Figure 12: 4 classes, 10 distributions per class - set of means = $\{0.1, \mathbf{0.3}, 0.6, 0.9\}$ - mean 0.3

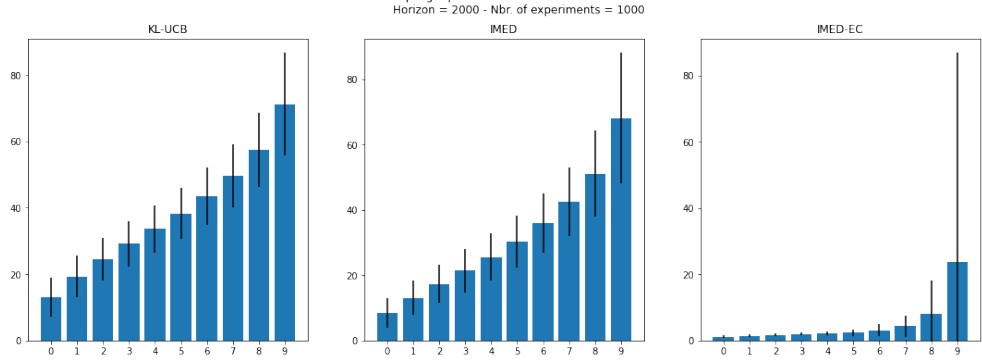

Figure 13: 4 classes, 10 distributions per class - set of means = $\{0.1, 0.3, \mathbf{0.6}, 0.9\}$ - mean 0.6

**Influence of the parameter $q$**  Here we show the numerical robustness of `IMED-EC` with respect to the lower bound parameter $q$ on the number of elements per classes. On the same bandit problem, we compare different instances of `IMED-EC` where different values of $q$ are used. In the legend, *opt.* stands for optimal and corresponds to the largest valid lower bound on the number of elements per class, *i.e.* the minimal number of elements in a class. The experiments done for Figure 15 are performed on a bandit problem with 4 classes and and 10 distributions per class. While $q$ increases up to the minimum cardinality of a class, we see that the performances of `IMED-EC` increases, which is expected. It is rather remarkable that once we go beyond that theoretical threshold, the performances of `IMED-EC` do not seem to deteriorate.

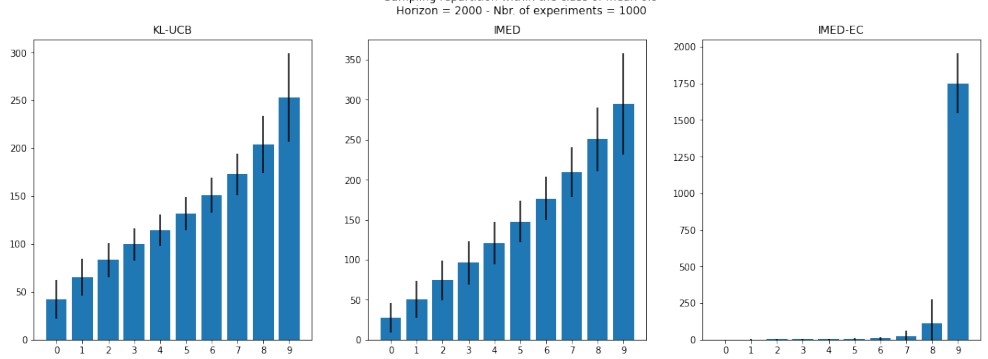

Figure 14: 4 classes, 10 distributions per class - set of means = $\{0.1, 0.3, 0.6, \mathbf{0.9}\}$ - mean 0.9

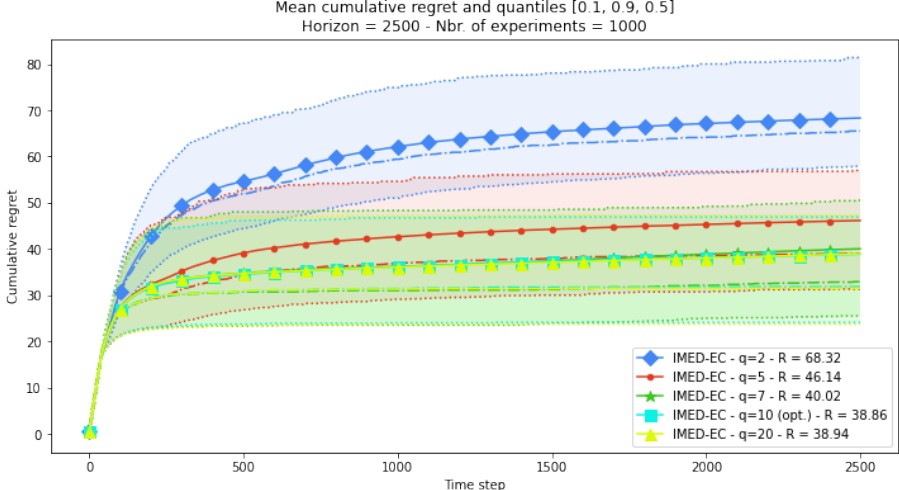

Figure 15: 4 classes, 10 distributions per class - set of means = $\{0.1, 0.3, 0.6, 0.9\}$

Finally, we explore the *dispatching* of `IMED-EC` and compare it to the one of `KLUCB` and `IMED` on this setting. We run the three algorithms 1000 times on a bandit problem whose number of class is 7 with means $\{0.1, 0.3, 0.4, 0.5, 0.6, 0.75, 0.9\}$ and an uneven number of Gaussian distributions with unit variance within each class. The chosen horizon is 2000. We assume that `IMED-EC` does not have perfect knowledge on the number of elements per class, and we use 3 as the lower bound parameter of `IMED-EC`. For some classes, we report the histogram accounting for the number of times distributions within each chosen class has been pulled. Error bars corresponds to the standard deviations and have been clipped to not go below the $x$-axis.

The comments that can be made about those plots are similar to the one that were already made for similar experiments. We included them to show that the behaviour of `IMED-EC` (and also the behaviour of `IMED` and `KLUCB`) is consistent across multiple settings. In particular, the algorithm `IMED-EC` exhibits the same aforementioned behaviour for the least suboptimal class, as it can be seen by comparing Figure 18 to the corresponding Figure 8 and Figure 13.

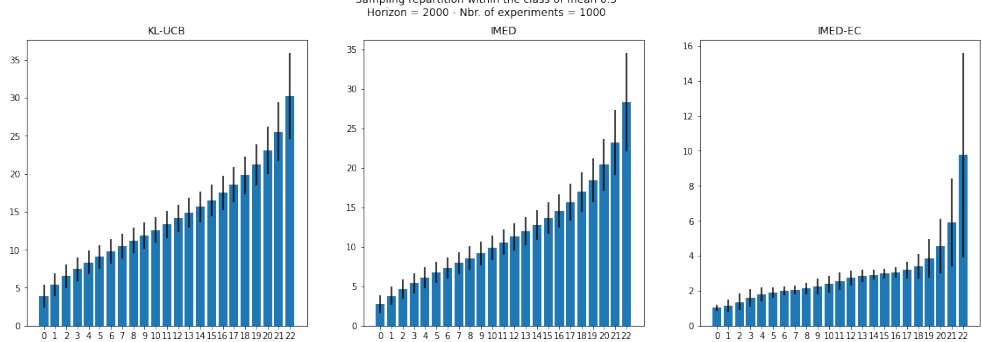

Figure 16: 7 classes - unbalanced - set of means = $\{0.1, \mathbf{0.3}, 0.4, 0.5, 0.6, 0.75, 0.9\}$ - mean $0.3$

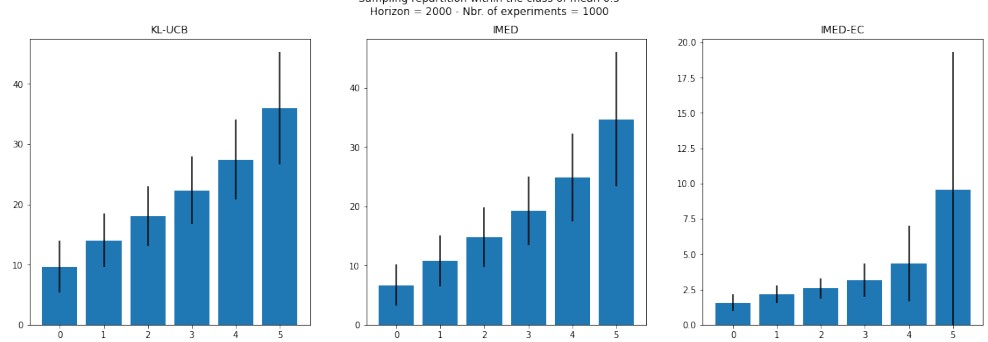

Figure 17: 7 classes - unbalanced - set of means = $\{0.1, 0.3, 0.4, \mathbf{0.5}, 0.6, 0.75, 0.9\}$ - mean $0.5$

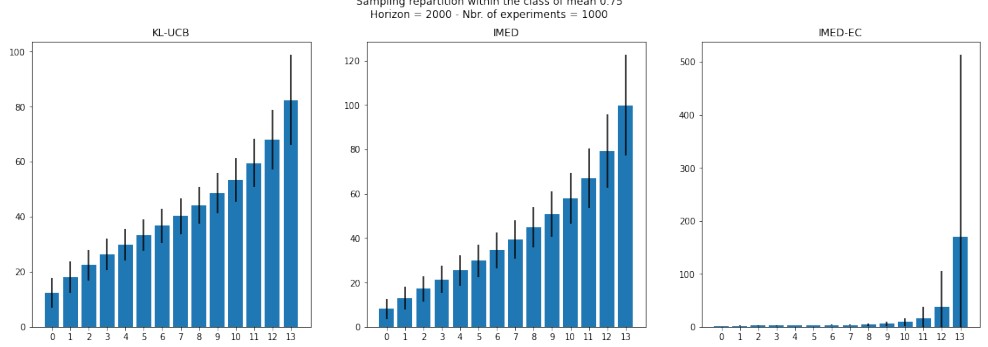

Figure 18: 7 classes - unbalanced - set of means = $\{0.1, 0.3, 0.4, 0.5, 0.6, \mathbf{0.75}, 0.9\}$ - mean $0.75$

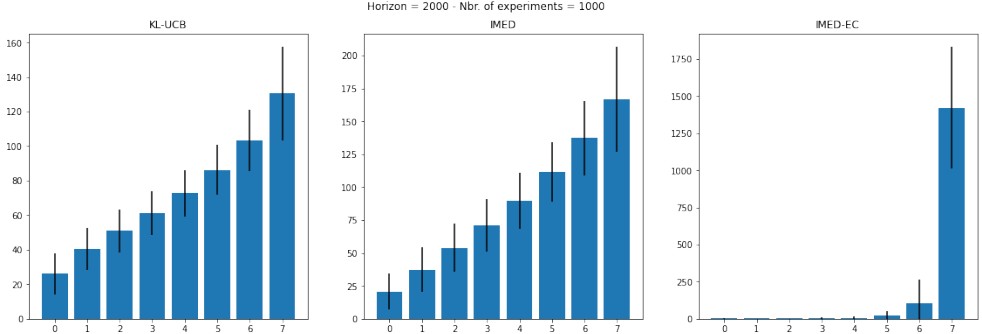

Figure 19: 7 classes - unbalanced - set of means = $\{0.1, 0.3, 0.4, 0.5, 0.6, 0.75, \mathbf{0.9}\}$ - mean $0.9$