# OpenReview forum: "Stochastic bandits with groups of similar arms."
_NeurIPS.cc/2021/Conference — NeurIPS 2021 Poster_

### Official Review · Reviewer_nrkQ · 2021-07-02

**Rating:** 6
**Confidence:** 3

**Summary:**

The paper studies a version of the stochastic multi-armed bandit problem where arm-means have a multiplicity of at least $q$, meaning, for any given arm "a," there exist at least $q-1$ other (distinct) arms with a mean reward same as that of "a." The parameter $q$ is assumed known. The reward distributions are assumed to belong to a one-dimensional exponential family. Under this premise, the authors show refined lower bounds for the expected sampling times of inferior arms (also regret) that exhibit a $q^{-1}$-scaling. Additionally, leveraging the underlying problem structure, a refinement of the KL-divergence-based terms appearing in the lower bound is also shown. The authors propose a modification of the IMED algorithm that suffers a regret within a multiplicative "control" factor of the lower bound. All in all, the takeaway is that the achievable regret is reduced by a factor of $q$. The reported numerical experiments indicate robustness to misspecification of $q$.

**Limitations And Societal Impact:**

Unfortunately, the paper appears to have overloaded notation with plenty of typos and notational inconsistencies, which make it hard to parse. It could seriously benefit from a revision along these lines. Typos (not a comprehensive list) and some general remarks are noted below:

1. Line 51: I suggest you use a different notation for $\mathcal{B}_q(\mu)$, it gets really confusing later on when you overload it further.

2. Lines 134-137: Please rephrase.

3. Line 151: The mapping $\mu^\prime$ is not defined before. Also, I believe there is scope for making the definition of "confusing" instance, less-confusing (no pun intended).

4. Lines 158-163: Please decide upon a stable convention for KL divergences. As in, KL$(.,.)$ vs. KL$(.||.)$. Similarly for $\mathcal{K}_{eq}$.

5. Theorem 1 second line: A should be $\mathcal{A}$, equivalence class in $\nu$ (not $\mu$).

6. Line 165: What is a "bandit dependent lower bound?"

7. In equation (2), the $\nu_a$'s should be $\mu_a$'s (there are other instances too, not noted here). The $\mu$ below the "inf" should be $\nu$. Also, how is $\lambda$ specified?

8. It would greatly help the reader if you could provide intuition on the combinatorial vs. non-combinatorial regimes in terms of the $q,2q$ business.

9. Equation (5): $\nu_a$ should be $\mu_a$ and a possible summation missing?

10. Equation (6): Extraneous log term and unspecified $\lambda$.

11. Line 210: Typo in the range of summation. Also, the $X_{t}'s$ haven't been defined before.

12. Just below line 214: It should be $\mathcal{A}_{\ast}(t)$ below "min."

13. Please mention the acronym IMED-EC next to Algorithm 1.

14. Line 221: *makes.

Some of these issues recur subsequently too, I am not mentioning them again. Overall, I like the punchline of the paper -- a factor $q$ reduction in achievable regret if arm-means have $q$-multiplicities. But the paper needs a clean-up. I am open to increasing my score if aforementioned points are satisfactorily addressed.


**Main Review:**

UPDATE:
I thank the authors for their response; I have revised my rating to 6 since the authors could satisfactorily answer several of my questions. However, in light of the issues raised by the other reviewers, I am also minded to reduce my confidence score to 3.

-------------------------------------------------------------------------------------------


The model and results certainly seem interesting. Since multiplicity of arm-means is a key feature, a brief survey of extant works that have considered such models will only make the literature review more comprehensive. Recent papers that come to mind are https://arxiv.org/pdf/2105.10721.pdf and https://arxiv.org/pdf/2103.12452.pdf (check references for related works). Cited papers assume $q=\infty$, basically meaning that the arms are sampled from an infinite "reservoir." However, the reservoir is assumed to hold a fixed (positive) fraction of "optimal" arms which are "well-separated" in the mean-reward from other sub-optimal types. Despite an infinity of arms in said problem, the papers show that logarithmic regret is still achievable. It would be interesting to see how the finite-armed model with a multiplicity of arm-means studied in this paper relates to its infinite-armed counterpart from cited works, achievability of logarithmic regret surprisingly being the unifying thread.


**Time Spent Reviewing:**

12

---

> ### Author Response · Authors · 2021-08-09
> **Rebuttal for nrkQ**
>
> Thank you all for your detailed reviews and for carefully spotting the typos. We will make sure to correct them in the paper. Thank you as well for acknowledging the "punchline" of the paper. This will help us make the paper more impactful by adding such a line at the end of the introduction along with the experiment mentioned in the general rebuttal.
>
> ## Typos/Notations
> In particular, l. 151, $\mu'$ is the set of means associated to $\nu'$, similarly to what is done l. 29. We will make this clearer. Eq. 2 and 6, $\lambda$ should be $\mu_*$.
>
> However, we disagree on two points: Because of assumption 2, one could change $\nu_a$ for $\mu_a$ in some part of the paper. However, we use this assumption only for the regret upper bound, it is not used for the lower bound.
> We think the confusion comes from the phrasing of assumption 2. We will change the l. 163 to:
>
>  $$
> \forall \kappa , \chi \in \mathcal{F} , KL( \kappa \| \chi ) = \tilde{KL} ( \mathbb{E}\_{ X \sim \kappa} (X) \| \mathbb{E}_{X \sim \chi} (X) )
> $$
>
> ($\tilde{KL}$ is the parameterization of the $KL$ by the means).
>
> The second point is about the $X_t$'s. They are introduced l. 33 of the paper. However, we agree that the first time this notation is reused (l. 210) is far from the introduction and we will remind the notation just after the formula (l. 211):
> $\hat\mu_a(t)=\frac{1}{N_a(t)}\sum_{s=1}^t X_s1\{a_s=a\}$, **where $X_s$ is the sample collected by the algorithm at time step $s$**.
>
> By "bandit dependent lower bound", we meant a "problem dependent lower bound" that depends on the considered bandit instance. The purpose is to inform the reader that we study a problem dependent lower bound rather than a worst case bound as it is also found in the bandit literature. For instance, see Lattimore et al. (https://arxiv.org/pdf/1411.2919.pdf) and Agrawal et al. (https://arxiv.org/pdf/1209.3353.pdf). We will fix the wording.
>
> ## Confusing instance
> The (known) notion of confusing instance is introduced in def. 3 with a lot of notations (the $\mathcal{B}_q$s) and we thank you for pointing out that, together, this might be unclear. We will use different notations for the set of confusing bandits, and for the set of confusing means and avoid overloading the same notation.
> To improve clarity and link theorems to intuitions, we will emphasize the notion of *confusing instance* and provide more explaination about it. Better explaining this notion will help give intuition and link sections of the paper together. However, the notion of confusing instance is already known (sometimes referred to as confusing parameters) and used in the community that is interested in problem dependent lower bounds in bandit settings, which is why we did not spend too much time on it. For instance, it is used in the recent papers of Degenne et al. (https://arxiv.org/pdf/2007.00969.pdf) and Magureanu et al. (https://arxiv.org/pdf/1405.4758.pdf). We understand that this notion might not be as standard as we thought and will add a small section in the appendix to recall it more precisely and clarify how it is used (before the main proofs).
>
> Please refer to the section **rephrasing** (confusing instance, combinatorial and non-combinatorial regime) of the **rebuttal addressed the reviewer KxtC** to see how we can better explain confusing instances and the two regimes.
>
> ## Complexity (rephrasing)
> At each time step $t$, IMED-EC computes the IMED indexes $I_a(t)$ of all the arms. IMED algorithm chooses the next arm to pull as an arm having the minimal IMED index. The algorithmic complexity of IMED is therefore, at each time step, $|\mathcal{A}|$ times the complexity of computing an index. On the other hand, IMED-EC chooses the next arm based on $I_q(t)$ and $I_*(t)$. To compute $I_q(t)$, one has to find a set of arms having the $q$ smallest IMED indexes and compute the sum of indexes on this set. Finding the $q$ smallest elements can be done by maintaining a sorted array of IMED indexes while computing them. The procedure costs a constant factor of $\log|\mathcal{A}|$. Computing $\mathcal{A}_*(t)$ can be done by maintaining a set of arms having the best empirical mean (adds a constant factor). Therefore, the complexity of choosing the next arm with IMED-EC is no more than $\log|\mathcal{A}|$ times the complexity of choosing the next arm with IMED.

---

### Official Review · Reviewer_dE1k · 2021-07-13

**Rating:** 5
**Confidence:** 4

**Summary:**

The paper extends the study of multi-armed bandits to the case where there are groups of arms having the same distribution (in the exponential family), and a lower bound $q$ on the size of these groups is known to the learner. The authors introduce a new parametric asymptotic lower bound for this setting. They build on the IMED strategy from Honda and Takemura to propose an adapted strategy. This new strategy has a regret that is at worst 2 times the asymptotic lower bound, and q-times smaller than the regret of an unstructured bandit instance having the same number of arms.

**Limitations And Societal Impact:**

I am satisfied with the way the authors addressed the limitations of their work.

**Main Review:**

## Originality / Significance
 The authors consider a setting where there are groups of arms having the same distribution and a lower bound $q$ on the size of these groups is known to the learner. They justify the interest of this setting by an example of problem where each arm can be described with a list of categorical attributes, and the reward function only depends on a subset of them. This is an original problem, that has not yet been studied, as I understand. The lower bound is original and even though the algorithm heavily resembles another algorithm( IMED from Honda and Takemura), the algorithm is efficient and the upper bound is technically involved.

However, a first question arises regarding the interest and significance of this setting. I would like to understand from the authors what are the practical cases where the learner knows a lower bound on the size of the groups but does not know at least a subset of the features that are redundant and hence the groups? The authors attempt to explain it line 62, but do not succeed in making it clear, in my opinion.

## Clarity/ Soundness
The paper lacks precision on multiple occasions, which harms its clarity but also makes the reader doubt its soundness :
- In Theorem 1, should $\lambda$ be replaced with $\mu^*$? Otherwise, how is it defined? (same for l.174 and equation 6)
- Why does $|c_q|\geq 2q$ belong to the non-combinatorial regime? There is absolutely no explanation for this fact.
- The paragraph starting in l186 and explaining the difficulties with the combinatorial regime is not clear. Can the authors reformulate the paragraph?
- In Theorem 3, is the liminf a limsup? A liminf would not be satisfactory.

## Minor flaws
- In the proof of Theorem 1, the use of the Fano inequality should be explicit, even if it is classic.
- Also, the authors should explain how they swap the liminf and the min , line 493.
- The fact that the distributions are parametrized by their means because of assumption 2 should be explained earlier than on page 4.  Otherwise, the equivalence classes based on the means only seem strange.

## Typos
- l45 : we study/studies
- l114 : The authors show
- l170: combinatorial
- l175: amounts
- l178: assumption
- l179: times
- l221: makes
- l246: assumptions
- l252: a striking result





**Time Spent Reviewing:**

6

---

> ### Author Response · Authors · 2021-08-09
> **Rebuttal for dE1k**
>
> Thank you all for your detailed reviews and for carefully spotting the typos. We will make sure to correct them in the paper. Thank you as well for acknowledging the originality of our work. Regarding the resemblance to IMED, we would like to point out that, from our perspective, it is a strength. You may want to read the **Algorithm: a paper's strength** section of the rebuttal addressed to reviewer VZyd for further clarification about this.
>
> # Setting
> A general rebuttal has been written to tackle the issue of the setting. We hope this clarify the questions you rose in the significance part of the review.
>
> # Clarity
> ## Typos/clarification
> You are right that $\lambda$ should be replaced by $\mu_*$ in the Theorem 1. Also, $\liminf$ should be a $\limsup$ in the Theorem 3, good catch! Thank you for your careful reading, we will make sure to correct these silly typos in the paper.
> In the proof of Thm. 1, we mainly use the data processing inequality (as in Garivier et al. https://arxiv.org/pdf/1602.07182.pdf) which is indeed similar to the Fano's inequality while not quite the same. This is why we did not mention it in the proof. To avoid heavy notation we did not include a trailing $\min$ at each line, and swapping the $\min$ and $\liminf$ can be done this way. To avoid confusion, we will add one line in the proof to explain this while keeping the notation less heavy.
>
> ## Mean-structure
> As explained in the general rebuttal, the reason why assumption 2 is stated on p.4 is because we don't need it to define an equivalence structure by the means since the regret (def. 1, l. 41) is defined as a function of the means,
> $$ \mathcal{R}( \nu,T)=\sum_{a\in\mathcal{A}}(\mu_* -\mu_a)\mathbb{E}_\nu N_a (T) .$$
>
> In particular, if we were to know the set $\mathcal{C}$ of classes, we could sum on the classes $c\in\mathcal{C}$ rather than arms,
> $$ \mathcal{R}(\nu,T)=\sum_{c\in\mathcal{C}}(\mu_* -\mu_c)\mathbb{E}_\nu N_c (T) ,$$
>
> where $\mu_c$ is the mean of arms in class $c$ and $N_c(T) = \sum_{a\in c} N_a(T)$ is the number of times arms in set $c$ have been sampled up to time T.
>
> Furthermore, the structured lower bound does not need assumption 2 because it uses the $\mathcal{K}_\mathcal{F}$
>
> and $\mathcal{K}_{eq}$ functions. This is very similar to the unstructured setting, see eq. 1, l. 41., in which the lower bound of Burnetas and Katehakis does not need such an assumption (and only uses the $\mathcal{K}_\mathcal{F}$ function).
>
> We hope this clarify why assumption 2 is only introduced on p. 4.
>
> ## The two regimes
> Please refer to the section **confusing instance** of the **rebuttal addressed the reviewer KxtC** to see how we can better explain confusing instances and the two regimes. In particular, we rephrase in the subsection **non-combinatorial regime**, the two reasons why $|c|\geq 2q$ does not belong to the combinatorial regime (reminder larger than $q$ and linearity of the sum). The paragraph starting l. 186 is rephrased in the subsection **combinatorial regime**. Thank for pointing out that this might be unclear to some readers.

---

### Official Review · Reviewer_KxtC · 2021-07-15

**Rating:** 7
**Confidence:** 3

**Summary:**

This paper considers a new variant of the classic stochastic bandit problem, analysing the complexity of the problem and proposing an effective algorithm. The new problem matches the classic problem definition of the K-armed bandit, except all arms are grouped into classes with identical distributions. Each class contains at least q arms, and the decision-maker is assumed to know q a priori but not the classes. A motivating scenario where this may arise is one where arms have multiple categorical features but certain features are uniformly redundant in the sense that the reward distributions do not vary with respect to these features. The decision-maker would in such a setting know (a lower bound upon) the number of redundant features but not which features are redundant (and as such which arms are equivalent).

The authors derive an asymptotic lower bound on the number of plays of suboptimal arms, and then propose an algorithm based on the IMED indices of Honda and Takemura (2015). The regret of this algorithm is shown to coincide with the lower bound in certain settings -  where classes are equally sized (roughly speaking) – and differ from the lower bound by no more than a constant factor in other settings. The algorithm is shown to outperform non-bespoke approaches which treat are unaware of the arm class structure, and have some robustness to misspecification of q.


**Limitations And Societal Impact:**

They have adequately addressed limitations

**Main Review:**

The paper is generally of a high quality. The theoretical underpinnings are strong, the authors write mostly very well, and the solution to the problem is appropriate and builds nicely on existing work. I have two main issues with the presentation of the material – firstly that the more theoretical aspects could be explained more clearly in the main text (see section ‘Clarity’) and secondly that the motivation for considering the problem could be more detailed and have a potential issue resolved (see section ‘Significance’). I am impressed by the work and would be happy for the paper to be accepted as is, but could also give a yet higher score and feel the paper would be genuinely improved is these issues were addressed.

ORIGINALITY AND QUALITY: My view is that the paper clears the acceptance bar in this regard. The problem is interesting and the work on it is new, accurate as far as I can tell (I have not been able to thoroughly inspect the appendices in the time-limited reviewing window) – certainly the main results make sense and have required some effort to derive. The performance of the algorithm is impressive and the algorithm is simple to implement and computationally efficient.

CLARITY: I really enjoyed the introductory section of the paper. The writing is excellent – the authors successfully adopt a blend of conciseness, clarity, and accuracy that explains core concepts quickly but accessibly. This is genuinely some of the best scene-setting I have seen in a bandit theory paper while reviewing at ML conferences. In Section 2, I feel there is something of a departure from this style as the mathematical material becomes more complex. I feel that this is not necessary. There is room within the page limit to guide the reader with more context and intuition here. The definitions and assumptions, and indeed Theorems/Lemmas to some extent, are presented without much linking text or context. The intuitions behind the lower bounds are complex but not (I feel) unaccessible to the unfamiliar reader, and as such I feel more could be done to explain the various steps here. As examples, I would explain to the reader the role of the ‘confusing instances’ (to find the exploration needed in the most challenging cases), give more intuition as to where the various terms in Theorem 1 arise from, percolate the discussion of non-combinatorial and combinatorial regimes with plain text explanations of the concepts (e.g. ‘an instance where all groups of similar of arms are of the same size’), and being more clear in the first instance what the remainder of the section aims to do to improve upon the insight given by Theorem 1. An explanation of how such improvements could be made via the rebuttal would lead me to increase my score.

SIGNIFICANCE: The problem is interesting theoretically, and I could imagine the paper laying the foundation for the study of more complex variants (contextual, combinatorial, non-stationary etc.) which may have genuine applications. The mention of categorical features suggests a setting in which this problem would genuinely arise, however the motivation in this regard is scant. I accept that this predominantly a ‘theory paper’ but I do think the paper could be improved with a more convincing argument as to the motivating applications of the work. If they are genuine – why not explain/cite in more detail? Further, why is it preferable to approach this redundant categorical feature problem via the  approach in this paper over a more structured model selection-type approach where we can use the information that arms which correspond to matching levels of more categorical features are more likely to be part of the same class? I would like to see the motivation for the work addressed in more detail via the rebuttal.

Finally, some minor typographical errors as follows: in line 114, should be ‘authors’ not ‘author’ in line 126, should be ‘paper’ not ‘papers’, in Thm 1, should be ‘suboptimal arms’ not ‘suboptimal arm’ there are further similar issues around pluralisation throughout the remainder - in line 179, should be ‘times’ not ‘time’ in line 180 the l in ‘Lemma 1’ should be capitalised, in line 281, should be ‘imperfect’ not ‘unperfect’. [NB, I have not considered these minor typos in assigning my review score].


**Time Spent Reviewing:**

5.5

---

> ### Author Response · Authors · 2021-08-09
> **Rebuttal for KxtC**
>
> We would like to thank you for your enthusiasm and for highlighting the strengths of the paper. Thanks to your review, we will broaden the introduction to emphasize the problem we study and clarify its relationship to other bandit settings.
>
> # Setting (Significance)
> A general rebuttal has been written to tackle the issue of the setting. We hope this clarifies the questions you rose in the significance part of the review.
>
> # Clarity
> To improve clarity and link theorems to intuitions, we will emphasize the notion of *confusing instance* and provide more explanation about it. Better explaining this notion will help the intuition and link sections of the paper together. However, the notion of confusing instance is already known (sometimes referred to as confusing parameters) and used in the community that is interested in problem dependent lower bounds in bandit settings. This is why we did not spend too much time on it. For instance, it is used in the recent papers of Degenne et al. (https://arxiv.org/pdf/2007.00969.pdf) and Magureanu et al. (https://arxiv.org/pdf/1405.4758.pdf). We understand that this notion might not be as standard as we thought and will add a section in the appendix to recall it more precisely and clarify how it is used (before the main proofs).
>
> ## Rephrasing
> ### Confusing instance
> *Before definition 3, l. 148* - Most confusing instances allow to assess the *intrinsic difficulty* of a bandit problem and allows to compute lower bounds on the number of times suboptimal arms are pulled. The lower bounds informs us on the minimal amount of exploration one need to do to solve a bandit problem. More formally, a *confusing instance* $\nu'$ associated to a suboptimal arm $a$ for a bandit problem $\nu$ is a bandit instance with the same set of arms as the original one but in which $\mu_a$ has been changed to $\mu_a'>\mu_*$. An optimal sampling strategy (one that does not sample suboptimal arms too much) should behave differently on the two problems. Studying this difference, we can compute the minimal amount of exploration performed by an optimal strategy on arm $a$ in the original problem $\nu$. Doing so for all suboptimal arms allows to bound the number of samples of suboptimal arms and therefore characterize the intrinsic complexity of a bandit instance $\nu$.
>
> In a structured setting, a confusing instance also has to respect the structure. In our case, it means that a confusing instance cannot have a class with less than $q$ arms. We will therefore consider confusing instances associated to classes rather than individual arms.
>
> ### Insight on Thm. 1
> *l. 168* - The proof of Thm. 1 (in appendix) considers confusing instances in which $q$ arms from a suboptimal class $c$ are moved above the optimal one (w.r.t. the mean). If there are $q$ arms in the class, then there are no remaining arm to move. If there are more than $2q$ arms, then moving $q$ arms creates a reminder of size larger than $q$ meaning that the crafted confusing instance respects the equivalence structure. However, if there are between $q+1$ and $2q-1$ arms, then the reminder is of size larger than 1 but strictly smaller than $q$. The created confusing instance does not respect the equivalence structure and we have to deal with the arms in the reminder. There are $|c|$ choose $q$ possible choices to move $q$ arms from class $c$.  All in all, the lower bound involves a combinatorial optimization problem as shown in Thm. 1.
>
> ### Non-combinatorial regime (rephrasing)
> *l. 171* - If $|c|\geq2q$, the problem in eq. 2 is no more a combinatorial optimization problem for two reasons. First, when crafting a confusing instance, the reminder is of size larger than $q$ and the infimum from Thm. 1 disappears. Indeed, the infimum is always 0 as this quantity can be obtained by choosing $\mu_a'=\mu_a$ for all $a\in c\setminus c_q$ (notation of the Thm. 1). Second, the minimization over all the $|c|$ choose $q$ partitions of the class $c$ involves the sum of $q$ quantities (l. 172), hence is not a combinatorial problem anymore. Indeed, the computed quantity is linear, and we can sort all the costs,
>
>  $( \mathbb{E} (N_a)\mathcal{K_F} $  $( \nu_a\ \| \mu_* ) )_a$, and sum the $q$ smallest.
>
> ### Combinatorial regime (rephrasing)
> *l. 186* - If the suboptimal class $c$ is such that $q<|c|<2q$, then the reminder is such that $0<|c\setminus c_q|<q$ and the infimum in Thm. 1 is not 0 (recall that here the reminder is not empty). Since the infimum is not a linear operator and is a priori different for all choices of $c_q$, the minimum over all the $|c|$ choose $q$ partitions of the class $c$ cannot be simplified, hence a combinatorial optimization problem.
>
> We hope this clarify the notion of confusing instance and how it is linked to the $\leq 2q$ vs. $>2q$ regimes and show how precise modifications could be implemented.

---

> > ### Comment · Reviewer_KxtC · 2021-09-02
> > **Maintaining my score**
> >
> > Hi authors,
> >
> > Thank you for your replies to my comments. I still consider this to be a strong submission and will keep my score

---

### Official Review · Reviewer_VZyd · 2021-07-16

**Rating:** 5
**Confidence:** 4

**Summary:**

In this work the authors present an algorithm for the stochastic multi-armed bandits setting where it is known a priori that arms come in group of size at least q, all having the same mean. A lower bound on the regret is given, as well as an asymptotically order-optimal achievability scheme.

**Limitations And Societal Impact:**

There are no negative societal impacts in this theoretical paper that focuses on regret minimization in the stochastic multi-armed bandit setting.


**Main Review:**

1. The motivation provided for this problem is relatively weak. The primary example the authors use to motivate this setting involves arms with differing values across several categorical attributes (some of which are redundant), which has significantly more structure than just groups of arms with the same means. This is more reminiscent of Factored Bandits [Zimmert and Seldin, NeurIPS 2018], which the authors should discuss and compare against.
2. Lack of clarity in simulations; in the paper there are several errors regarding simulations, e.g. for Figure 1 it claims 3 sets of means but lists a set of 4. Similarly in Figures 10,16,17,18,19 it is claimed that there are 4 classes, but 7 means are given. More importantly however, the authors poorly document their simulations, so it is unclear from the paper the true impact of the robustness of the algorithm to the parameter q. For example, it is only given for Fig 3 that there are 7 classes with given means, with an uneven number of Gaussian distributions in each class ranging from 4 to 23. It is not given which class has which size, let alone the total number of arms. For example, if the best class was the one with cardinality 4, this would seem to be a harder problem instance for the algorithm than if the class with cardinality 4 constituted the worst class. As such, the reproducibility and significance of Figure 3 are unclear. Critically, one of the two uses of q is constructing A_*(t), and so the relevant test regarding the robustness of the algorithm to q would seem to be when A_*(t) is impacted, yielding fewer than q optimal arms (which would appear to lead to linear regret, as suboptimal arms are included in A_*(t) ).
3. The lower bound appears to be novel, but the algorithm itself seems to be only a minor modification of the original IMED.
4. There does not appear to be any characterization or bounding of alpha or f in Theorem 2 in the main text. While the empirical performance of this algorithm seems promising, a discussion on alpha and f and the possibility for finite sample guarantees would strengthen this paper.

Minor comments:
- There were many typos and grammatical issues in this paper, which detract from the overall work. Some are listed below
- Title should not have a period at the end
- Line 40 that we assume ‘is’ unknown to the learner
- Lines 55/56 arms should be plural
- line 127 ‘makes appear’
- line 135 duplicated text
- Line 138 “2 time from” -> a factor of 2
- Line 146 have->has
- Line 210 summation should start from time index 1, and t is overloaded as both the time index and the dummy variable in the sum
- line 266 annex->appendix
- Line 272 duplicated compare
- line 448 Monotony of the KL -> monotonicity

**After Rebuttal:**
The authors clarified some helpful points in the rebuttal:

Experiments: Including all relevant experimental details in the Appendix would help contextualize the simulations; a reader should not need to look at the raw code in order to understand the details of the experiments.

Robustness: Thank you for clarifying $A_{*}(t)$. However, it is still unclear to me if Figure 3 shows robustness of this scheme with respect to q. From my understanding looking at the experiments, the authors simply vary the number of arms in a suboptimal class, where the interesting experiment to run would be seeing how the algorithm performs if there are fewer than q optimal arms (to which Figure 15 offers some partial insight).

$\alpha$ and $f$: Bounds and / or considerable discussion on these factors would greatly strengthen this work, as these quantities appear in the primary Theorem of this paper. If it is too difficult to provide these in general, specializing to Gaussian or Bernoulli instances would still be extremely helpful, to give a sense of how these quantities can be expected to scale. It was difficult for me to follow the appendix references as the analysis is not very self contained, and it is not clear why $f$ in particular goes to 0 in the limit.

This paper has its upsides, i.e. the problem is fundamental (equivalence classes of means) even if the motivating example is not well thought out, and the technical contributions (primarily the lower bound) are interesting and novel. However, the additional discussion does not change my score; the vague experiments and general lack of care taken with the writing are concerning. To address these issues, the authors will have to go through and correct the individual points we have listed, and all the others we did not. In its current form the presentation is poor so I cannot recommend acceptance. However, I will not object to acceptance either, so I retain my score of 5.

**Time Spent Reviewing:**

4.5

---

> ### Author Response · Authors · 2021-08-09
> **Rebuttal for VZyd**
>
> # Setting
> Regarding the motivation of the setting, please refer to the general rebuttal. Thanks to your comment, we will add the reference you mentioned (factored bandit) in the literature review in order to better emphasize the difference with our setting.
>
> # Experiments
> We would like to thank you for your careful reading regarding the experiments and Figures. We understand that the multiple typos in the figure's titles might have hindered the comprehension. The typos that you mentioned will be fixed. However, we would like to point out that python code to reproduce the figures was given in the supplementary material. In particular, Figure 3 can be reproduced by executing the cells below *XP 5* of the *regret_experiment.ipynb* file.
>
> ## Robustness
> Our IMED-EC algorithm takes **one hyper-parameter** as an input, which is **a lower bound q** on the number of arms within each class. The first parts of the paper prove that we can control the regret incurred by IMED-EC if the hyper-parameter *q* indeed is a lower bound on the number of arms within each class. Lets call $M$ the smallest number of arms in a class. We know that IMED-EC has logarithmic regret whenever $1 \leq q \leq M$. The asymptotic regret bound decreases as $q$ increases from 1 to $M$. Because IMED-EC matches its own asymptotic regret bound, this fact appears in the experiments which is why we did not emphasize it much. To assess the robustness of IMED-EC to misspecification of $M$, we used values of $q$ larger than $M$, $q > M$. Figure 3 is an illustration of what happens to the empirical regret, in terms of expected value and quantiles. As it is written at the beginning of the experiment section, *graphs are representative of all the experiments that we conducted*, which is why we did not think it was important to provide the cardinality of the classes. This will be amended.
>
> ## The set $\mathcal{A}_*(t)$
> $\mathcal{A}_*(t)$ is the set of arms that have the best empirical mean at time $t$,
>
> $\mathcal{A}_*(t)=argmax_a\hat\mu_a (t)$.
> Hence it does not directly involve the parameter $q$. Note that when dealing with distributions having a density with respect to the Lebesgue measure (e.g. Gaussian distributions), this set only has one arm with probability one (after all arms have been sampled once).
> While we never mention such a set, we could be interested in the set of the $q$ best empirical arms. Maybe this is what you were referring to. Our IMED-EC algorithm never estimates the classes (or sub-classes of size $q$) of the problem nor aggregates the means/samples of several arms. This behavior of IMED-EC can be seen in the dispatching experiments.
>
> If by robustness, you mean the case when $q > | \mathcal{A}_* |$,
>
> ($\mathcal{A}_*$ the true set of optimal arms) then an experiment showing such a robust behaviour of IMED-EC is found on Fig. 15 (l. 714, yellow curve).
> We hope that this was the meaning of your question regarding the robustness. We are open to the discussion and will provide any clarification that might be asked (discussion are open on openreview after the rebuttal).
>
> # Algorithm: a paper's strength
> While the algorithm appears like a minor modification of the original IMED algorithm, this modification is theoretically not trivial to prove correct. We think that the simplicity of the algorithm is a strength: The lower bound on the regret involves a combinatorial optimization problem. An optimal algorithm would have to solve such a problem at each round and would therefore be highly inefficient. IMED-EC is based on a relaxation of the combinatorial optimization problem and proving its regret was non-trivial. The techniques used in the appendix to prove the effectiveness of this algorithm and bound the discrepancy with respect to an algorithm that would solve the optimization problem might also be useful to others. This will be emphasized in the camera-ready version.
>
> # $\alpha$ and $f$
> The $\alpha$ and $f$ functions are mostly used for deriving theoretical guarantees in IMED-EC regret analysis. $\alpha$ is controlled thanks to property 2 (l. 452) as in the IMED paper of Honda and Takemura. A finite sample analysis can be derived from a careful analysis of the term $f$. Indeed, $f$ makes appear (cf. l. 640) an explicit sum $\sum_{n=1}^{\infty}\exp(-n\psi_a^\star(\epsilon))$ (l. 618), plus the quantity $D(\epsilon)$ (l. 637) appearing in the finite-time analysis of IMED. Being more precise requires scrutinizing the properties of the considered family. We agree to quickly elaborate on this point in the final version.

---

### Author Response · Authors · 2021-08-09
**General rebuttal**

Thank you all for your detailed reviews and for carefully spotting the typos. We will make sure to correct them and do a full clean up of the paper. Thank you as well for acknowledging the strengths of the article. Below, we mainly address your concern regarding the setting. We then provide specific answers to each reviewer separately.
# Significance/Setting
The example given in introduction is mainly to illustrate a situation when there are naturally many arms many of which have same means. In this example, one may also have a good proxy for q, which is why we find it is an interesting example. In our view, this is an example, out of many other settings, displaying an equivalence structure. It therefore gives a bound on what is achievable by all the more structured settings (e.g. factored) displaying a similar equivalence structure. Our goal is not to solve the very specific structure that is considered in the introduction, but to target the challenges of handling mean-equivalence in general. Our findings reveal that dealing with (pure) mean-equivalence structure is already not trivial, due to the presence of combinatorial terms and yet there exists also in this case an efficient strategy, whose regret is controlled in a very precise way on the theoretical side, plus enjoys competitive numerical results.

The aim of the paper is to study a simple equivalence relationship, yet not trivial to analyze. We feel it makes sense to focus solely on an equivalence on the **means** as the regret (l. 41) only involves the means of the distributions. The specific details of the distribution being implicit in $\mathbb{E}_\nu$.
If we were to know the set of classes $\mathcal{C}$, then the problem would reduce to a bandit problem with $|\mathcal{C}|$ arms. By definition of $q$, we know that $|\mathcal{C}|\leq\frac{K}{q}$, where $K$ is the number of arms. In this paper, we show that without knowing the classes, we can achieve the regret of a problem having $\frac{K}{q}$ arms. If the classes $c\in\mathcal{C}$ all have the same cardinality $q$, $|c|=q$, then IMED-EC achieves the same regret upper bound than an algorithm knowing the classes (but not their means). We think that this is a striking result worth being advertised. We will add an experiment in the appendix that compares the regret incurred by IMED on the subset of arms consisting of one arm per class ($\frac{K}{q}$ arms in total) to the regret incurred by IMED-EC on the full set ($K$ arms in total) where $q$ is known. This will emphasize the similar performances and effectiveness of IMED-EC. We will mention this experiment in the main text.

Furthermore, while one could have thought that **estimating** the classes would be necessary (e.g. by aggregating samples), IMED-EC and the associated proof shows that this is not. **In this paper, we prove that exploiting a structure can be done without ever estimating it**. Indeed, IMED-EC never estimates the classes but rather exploits the fact that there are at least $q$ arms per class to reduce it exploration by a factor $q$. We think this is an interesting result worth being advertised as it may be useful in other settings.
# Example
Consider a bandit $\nu$ where arms $a\in\mathcal{A}$ are described by a list of $p$ categorical attributes $(z_1,\cdots,z_p)$. In this model, the expected reward of arm $a$, $\mu(a)$ is an unknown function of three attributes $(z_1,z_2,z_3)$, and can be written as $\mu(z_1,z_2,z_3)$. In that case, $q$ is linked to $p$ and the number of elements in each categorical attributes (as described in the main text). In an online and active setting, one could try to cluster and learn the relevant attributes while performing regret minimization.
We agree that this setting is more structured than the one we studied. On the other hand, one can imagine that for privacy reason, the learner does not have access to the categorical attributes directly and can only choose an arm $a$ without knowing the specific attributes associated to it. In this case, the setting is the one described in our paper.
# Literature review
Thank to nrkQ for pointing out two references that we were unaware of and that we will include in the literature review since there are indeed similarities between the settings.
*Bandits with many optimal arms*: we will point out that the authors consider a situation in which there are several optimal arms. We will stress the difference that there is no notion of class nor constraints on the level-sets of the $\mu$ function.
*From Finite to Countable-Armed Bandits*: we will point out that the authors consider a finite set of types of arm, each characterized by their mean. This is similar to our setting in which we consider bandit with groups/types of arms. We will stress the difference that there is no constraints on the level-sets of the $\mu$ function and that the authors focus on developing new mathematical tools in the OFU paradigm while we focus on an IMED like approach.

---

### Decision · Program_Chairs · 2021-09-27

**Decision:**

Accept (Poster)

**Comment:**

The committee was divided about this paper: while the technical value is recognized by all and praised by one reviewer, it seems like at least two reviewers were puzzled by the clarity of the paper. I decided to read the draft myself to make my own opinion on whether the pass needed was minor (and does not require another cycle of review) or major. I think it is rather the former situation, though I see how a few typos at the wrong places may have mislead reader. Please update your work for the final version, removing typos and adding comments on motivation as well as citations to further highlight the connection with existing work and the potential impact of your discoveries.